# ECA: Efficient Continual Alignment for Open-Ended Image-to-Text Generation.

## Abstract

Incremental Learning (IL) for Open-ended Image-to-Text Generation (OpenITG) enables models to continuously generate accurate, contextually relevant text for new images while preserving previously acquired knowledge. Unlike prior studies, this paper addresses a more practical scenario in which the predominant category of visual data shifts over time as environments evolve. In this context, we introduce a new notion of continual alignment, which incrementally adapts the alignment module within pre-trained VLMs to preserve high-quality cross-modal representations. Based on this idea, we propose **E**fficient **C**ontinual **A**lignment (ECA), a novel exemplar-free IL approach for OpenITG. The key challenge is enabling the model to acquire new, task-specific features while minimizing interference with the established alignment without accessing raw data from previous tasks. To address this, ECA employs three core mechanisms: a **M**ixture **o**f **Q**uery (MoQ) module that adapts task-specific query tokens, a **Fi**sher **D**ynamic **Ex**pansion (FeDEx) that dynamically expands model structure based on a Fisher Information Matrix (FIM)-based metric, and an embedding dictionary with **D**ictionary **R**eplay (DR) to retain past knowledge. To evaluate ECA's performance, we construct four new IL OpenITG benchmarks that better reflect real-world scenarios. Experimental results demonstrate that ECA significantly mitigates catastrophic forgetting and improves IL performance compared to baseline methods. Benchmarks are available at `https://anonymous.4open.science/r/ECA-ToS-Benchmarks-FB17`.

## 1 Introduction

Open-ended Image-to-Text Generation (OpenITG) tasks, such as image captioning (Vinyals et al., 2015; Xu et al., 2015; Herdade et al., 2019; Ramos et al., 2023) and open-ended Visual Question Answering (VQA) (Antol et al., 2015; Xu et al., 2020; Fu et al., 2023), require Vision-Language Models (VLMs) to generate accurate, contextually relevant text based on given images. In real-world scenarios, the visual content shifts as environments and time evolve, leading to significant distribution changes. This dynamic setting motivates Incremental Learning (IL) for OpenITG, where a model adapts to evolving visual streams while maintaining generation quality.

As shown in Fig. 1(a), existing works (Del Chiaro et al., 2020; Zhang et al., 2023; Lei et al., 2023) often assume disjoint image categories or background scenes when splitting tasks, then train the model sequentially. However, this assumption does not always hold in practice because scenes contain multiple semantic elements whose prominence changes over time. For example, an indoor image dominated by "appliance" cues may later be dominated by "vehicle" cues, as illustrated in Fig. 1(b). Similarly, even if "vehicle" remains the dominant feature, the environment may introduce additional context. To capture these dynamic variations, we define an image's ***main topic*** as the semantic category of its most prominent object. Following the definition, we then split tasks by main topic to better reflect the continuous shifts of visual content. In our work, *we focus on IL for OpenITG tasks by adapting to the ever-changing main topics in visual data with overlapping semantics*.

Under our main-topic setting for IL in OpenITG, the key challenge is to maintain cross-modal alignment while countering catastrophic forgetting (McCloskey & Cohen, 1989) and interference from semantic overlap across tasks. As learning proceeds sequentially, the VLM tends to overwrite previously formed associations, which reduces the relevance and accuracy of generated text. Most

prior methods mitigate forgetting by fine-tuning fusion and language components sequentially with stored raw exemplars (Del Chiaro et al., 2020; Lei et al., 2023; Zhang et al., 2023). However, these approaches introduce three more major drawbacks. First, full-scale fine-tuning for large task-agnostic pre-trained models (Yuan et al., 2021; Zhang et al., 2022b; Chung et al., 2024) is inefficient and can erode pre-training gains (Zhai et al., 2023). Second, storing raw exemplars raises privacy and memory concerns. Moreover, since these methods are based on the assumption of disjoint distributions, they do not explicitly address semantic overlap across tasks, which is closer to real-world settings.

Motivated by these limitations, we first introduce a new notion of *continual alignment*, aiming to achieve the continual adaptation of the alignment module, which establishes the cross-modal alignment within pre-trained VLMs, to preserve high-quality cross-modal representations during sequential task learning. In our main topic setting, achieving continual alignment without saving raw exemplars necessitates addressing three challenges: **C1:** recurring semantics appear without task identifiers, which calls for the compositional reuse of earlier cues. **C2:** preserving established cross-modal alignment without saving raw exemplars is needed under distribution shift. **C3:** semantic overlap across tasks can trigger parameter conflict, which must be mitigated during adaptation. To tackle these challenges, we propose **E**fficient **C**ontinual **A**lignment (ECA), an exemplar-free framework operating at the alignment module. For **C1**, we introduce the Mixture of Query (MoQ) module, which learns task-specific query tokens and composes via attention to acquire new cues with minimal disruption to prior alignment. For **C2**, we design the Dictionary Replay (DR) module, which maintains a compact embedding dictionary to cap-

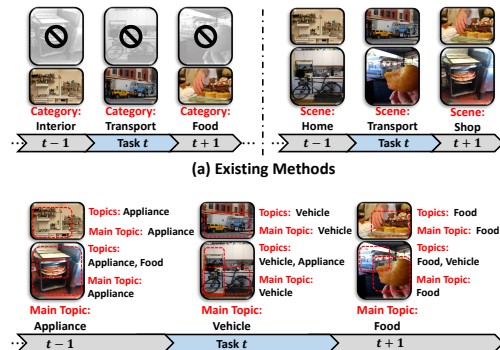

Figure 1: Comparison between (a) existing task splits (Del Chiaro et al., 2020; Zhang et al., 2023; Lei et al., 2023) and (b) our main topic split. **(a) top-left** illustrates methods that assume disjoint object categories and discard images containing multiple topics. **(a) top-right** illustrates methods that rely on disjoint background scenes. **(b)** defines each task by the image's dominant semantic category ("main topic"), which accommodates overlapping semantics and shifts in focus across time or environments, yielding a more realistic continual OpenITG setting.

ture task-agnostic visual components and replays them effectively without raw exemplars. To address **C3**, we propose the Fisher Dynamic Expansion (FeDEx), which computes a Fisher Information Matrix (FIM)–based metric, and selectively expands parallel adapters only when interference is detected, thereby preserving established alignment while allocating capacity to new topics. With the interplay of these three modules, ECA significantly alleviates catastrophic forgetting and enhances IL performance in OpenITG tasks. In this paper, we use BLIP-2 (Li et al., 2023) as a representative model to evaluate the ECA. BLIP-2 exposes the alignment module as a Q-Former, which connects a frozen visual encoder and a frozen language model, enabling us to isolate alignment module behavior and study continual alignment in a controlled evaluation setting. All baselines are trained under the same trainable scope to ensure a fair comparison.

To study the continual alignment and evaluate ECA, we build four IL benchmarks for OpenITG based on different main topics. We name these benchmarks as Topic of Semantic for COCO Caption (ToS-COCO Caption), ToS-VQAv2, ToS-TextCaps, and ToS-TextVQA, derived from MSCOCO ImageCaption (Lin et al., 2014), VQAv2 (Goyal et al., 2017), TextCaps (Sidorov et al., 2020), and TextVQA (Singh et al., 2019). These benchmarks cover image captioning and open-ended VQA. In our setting, models generate text across tasks without task-specific IDs.

In summary, **our contributions include:** 1) we propose ECA, a novel exemplar-free IL approach for OpenITG that updates only the alignment module while keeping heavy backbones frozen. To the best of our knowledge, we are the first to explicitly target preserving the cross-modal alignment of the alignment module in pre-trained VLMs during exemplar-free incremental learning for OpenITG. 2) we propose the Fisher Dynamic Expansion (FeDEx) with the Mixture of Query (MoQ) approach. These two modules adapt alignment module capacity and token composition to acquire task-specific features while preserving previously learned knowledge. By doing so, they continually adapt the alignment module in the pre-trained VLMs to maintain the cross-modal alignment. 3) we design a novel memory mechanism, Dictionary Replay (DR) based on sparse dictionary learning for IL

in OpenITG tasks instead of saving raw exemplars. 4) we construct four benchmarks i.e. ToS-COCO Caption, ToS-VQAv2, ToS-TextCaps, ToS-TextVQA, that closely mimic realistic scenarios by defining tasks according to image topics; and 5) Extensive experiments on our new benchmarks show that the proposed ECA achieves superior performance on IL for OpenITG tasks.

## 2 RELATED WORK

**Pre-trained Vision-Language Models.** Traditional VLMs for OpenITG perform full-scale end-to-end training (Herdade et al., 2019; Li et al., 2021; 2022; Wang et al., 2022d). However, with the rise of large-scale and task-agnostic pre-trained uni-modal models (Dosovitskiy et al., 2020; Brown et al., 2020; Chung et al., 2024), full fine-tuning becomes inefficient and inflexible. Recent VLMs (Alayrac et al., 2022; Li et al., 2023; Liu et al., 2023; Bai et al., 2025) adopt alignment modules to bridge frozen visual encoders and frozen Large Language Models (LLMs), realized as a Q-Former in BLIP-2–style models or as projector tokens in projector-based Multi-modal LLMs, which aligns visual features to the LLM token space and conditions the LLM on the image. However, the alignment module is sensitive to data distribution shifts and prone to catastrophic forgetting in IL scenarios (Zhao et al., 2024). Thus, we introduce the notion of *continual alignment* and then propose ECA to effectively improve the continual alignment ability of VLMs for OpenITG.

**Incremental Learning.** Incremental Learning (IL) aims to enable models to acquire new knowledge while preserving previous knowledge. Traditional IL methods generally fall into three categories: (i) regularization-based approaches (Li & Hoiem, 2017; Kirkpatrick et al., 2017; Aljundi et al., 2018; Ahn et al., 2019); (ii) rehearsal-based methods (Rebuffi et al., 2017; Douillard et al., 2020; Yan et al., 2021; Buzzega et al., 2020); and (iii) architectural-based methods (Fernando et al., 2017; Mallya & Lazebnik, 2018; Douillard et al., 2022; Wang et al., 2022b;a). Recent exemplar-free IL methods based on prompt-tuning (Wang et al., 2022c;e;f; Smith et al., 2023) have demonstrated strong performance on uni-modal tasks such as image classification, yet their effectiveness on multi-modal tasks like OpenITG remains under-explored. Recent IL approaches targeting OpenITG (captioning/VQA) tasks (Del Chiaro et al., 2020; Lei et al., 2023; Zhang et al., 2023; Chen et al., 2025) still face limitations. For example, VQACL (Zhang et al., 2023) combines prototype learning with exemplar buffers, posing privacy and memory issues. Moreover, these methods typically involve extensive fine-tuning of fusion and language components, which is inefficient for large-scale pre-trained models. More recently, several works on multimodal large language models introduce parameter-efficient fine-tuning (PEFT) approaches for *continual instruction tuning* (Chen et al., 2025; Cao et al., 2024; Zeng et al., 2025; Guo et al., 2025). Specifically, Continual LLaVA Cao et al. (2024) applies a low-rank embedding pool, CoIN (Chen et al., 2025) and HiDe-LLaVA Guo et al. (2025) leverage MoE-LoRA (Liu et al., 2024) in different ways, ModalPrompt Zeng et al. (2025) designs a dual-modality guided prompt tuning framework. However, these methods mainly aim at adapting models to evolving *textual instructions* rather than shifting *visual topics*. In contrast, our ECA approach effectively addresses these limitations by continuously adapting the alignment module of pre-trained VLMs without relying on exemplar buffers or task-specific identifiers, explicitly targeting IL scenarios involving continuous shifts in visual semantic topics.

**IL Benchmarks for OpenITG.** There are several IL benchmarks for OpenITG, including image captioning (Del Chiaro et al., 2020) and open-ended VQA (Zhang et al., 2023; Greco et al., 2019). In these works, tasks are typically split based on disjoint image categories. However, in real-world scenarios, a single image often contains multiple objects, and its overall semantics are better characterized by its dominant visual topic. Thus, the assumption of image distributions across tasks as disjoint is unrealistic. While similar work (Lei et al., 2023) splits tasks based on different scenes, focusing on the background, this approach fails to capture continuous shifts in dominant visual topics. In our work, we provide a new setting based on dominant visual topics and create four IL benchmarks for image captioning and open-ended VQA.

## 3 PROBLEM FORMULATION

This work aims to address the problem of continual alignment in pre-trained Vision-Language Models (VLMs) that incrementally align the cross-modal representations from new data distributions in two core OpenITG tasks: Image Captioning (IC) and Visual Question Answering (VQA).

Formally, the dataset for topic-$t$ (hereafter task-$t$) is $\mathcal{D}_t = \{(X_t, P_t, S_t)\}$, where $X_t = \{x_{t,i}\}_{i=1}^{n_t}$ denotes the set of input images, $P_t = \{p_{t,i}\}_{i=1}^{n_t}$ denotes the associated text inputs (prompts or questions), and $S_t = \{s_{t,i}\}_{i=1}^{n_t}$ contains ground truth sentences for each image-text pair, with $n_t$ as the number of instances in task-$t$. Given an image $x_{t,i}$ and the corresponding text $p_{t,i}$ from dataset $\mathcal{D}_t$, the VLM generates a predicted sentence, $\hat{s}_{t,i}$. Notations are detailed in Tab. 4 in Appendix A.

Recent VLMs (Alayrac et al., 2022; Li et al., 2023; Liu et al., 2023; Bai et al., 2025) are typically composed of three components: a visual encoder, an alignment module, and a Large Language Model (LLM). Accordingly, the sentence generation process at task-$t$ can be formalized as follows,

$$P(\hat{s}_{t,i}^j \mid \hat{s}_{t,i}^{<j}, x_{t,i}, p_{t,i}) = g_{\phi_t}\left(\hat{s}_{t,i}^{<j}, [A_{\omega_t}(f_{\theta_t}(x_{t,i})), p_{t,i}]\right), \qquad (1)$$

where $f_{\theta_t}(\cdot)$, $A_{\omega_t}(\cdot)$, and $g_{\phi_t}(\cdot, \cdot)$ denote the visual encoder, the alignment module, and the LLM within the VLM at task-$t$, respectively. To study continual alignment at the alignment module in isolation, we decompose a VLM as $(f_\theta, A_\omega, g_\phi)$, and regard BLIP-2 (Li et al., 2023) as a representative model to instantiate. Following the BLIP-2, we instantiate the alignment module $A_\omega$ with the Q-Former to bridge a frozen visual encoder and frozen LLM.

For BLIP-2, Eq. 1 can be reformalized as follows,

$$P(\hat{s}_{t,i}^j \mid \hat{s}_{t,i}^{<j}, x_{t,i}, p_{t,i}) = g_{\phi_\star}\left(\hat{s}_{t,i}^{<j}, [A_{\omega_t, Q_t}(f_{\theta_\star}(x_{t,i})), p_{t,i}]\right), \qquad (2)$$

where $\theta_\star$ and $\phi_\star$ are frozen across tasks. The alignment module $A_{\omega_t, Q_t}$ is the Q-Former at task-$t$, parameterized by $\omega_t$ and learnable query tokens $Q_t \in \mathbb{R}^{n_Q \times d_Q}$, where $n_Q$ is the number of tokens and $d_Q$ their dimension. Query outputs are linearly projected into the LLM space by a fully connected layer in $\omega_t$, enabling cross-modal alignment with frozen backbones.

Yet the learnable parameters $Q_t$ and $\omega_t$ are sensitive to distribution shifts, which causes catastrophic forgetting in IL. We therefore introduce a continual alignment mechanism to stabilize the alignment module across tasks, mitigating forgetting and supporting efficient adaptation to new data.

## 4 PROPOSED METHOD

### 4.1 OVERVIEW OF EFFICIENT CONTINUAL ALIGNMENT

To maintain accurate cross-modal alignment over sequential tasks, we propose Efficient Continual Alignment (ECA), an exemplar-free IL approach for OpenITG, and study it at the alignment module of BLIP-2. ECA comprises three components: ❶ **M**ixture **o**f **Q**uery (MoQ), ❷ **F**isher **D**ynamic **Ex**pansion (FeDEx), and ❸ embedding dictionary with **D**ictionary **R**eplay (DR). As shown in Fig. 2, a frozen visual encoder produces patch embeddings. MoQ learns task-specific query tokens and attentively aggregates them with the pretrained query tokens. The aggregated tokens are then passed to the alignment module (Q-Former) equipped with FeDEx. FeDEx selectively expands a parallel adapter based on an FIM-based metric, so new features are incorporated while preserving established alignment. Meanwhile, DR maintains an embedding dictionary and replays it during training to retain information from previous tasks. We detail each component below.

### 4.2 MIXTURE OF QUERY

**Motivation.** In VLM, the alignment module can leverage learnable query tokens $Q_t$ to expose visual evidence to the frozen LLM and establish the cross-modality alignment on the new task. In incremental learning, naively updating $Q_t$ based solely on the current task overwrites cues learned from past tasks, which leads to catastrophic forgetting. A straightforward solution is learning task-specific query tokens separately for each task and pick the appropriate set for a given image. However, since the visual inputs in OpenITG span a wide range of scenes and contexts, visual embeddings are widely dispersed and do not cluster into discrete categories. This dispersion makes direct identification of a single task-specific query token set impractical, which calls for a more flexible way to reuse and combine queries across tasks.

**MoQ.** To address this challenge, we propose a novel Mixture of Query (MoQ) module, which learns a unique set of task-specific query tokens for each task and uses an attention mechanism to dynamically aggregate them. Specifically, we first learn a set of task-specific query tokens $v_t \in \mathbb{R}^{n_Q \times d_Q}$ for each

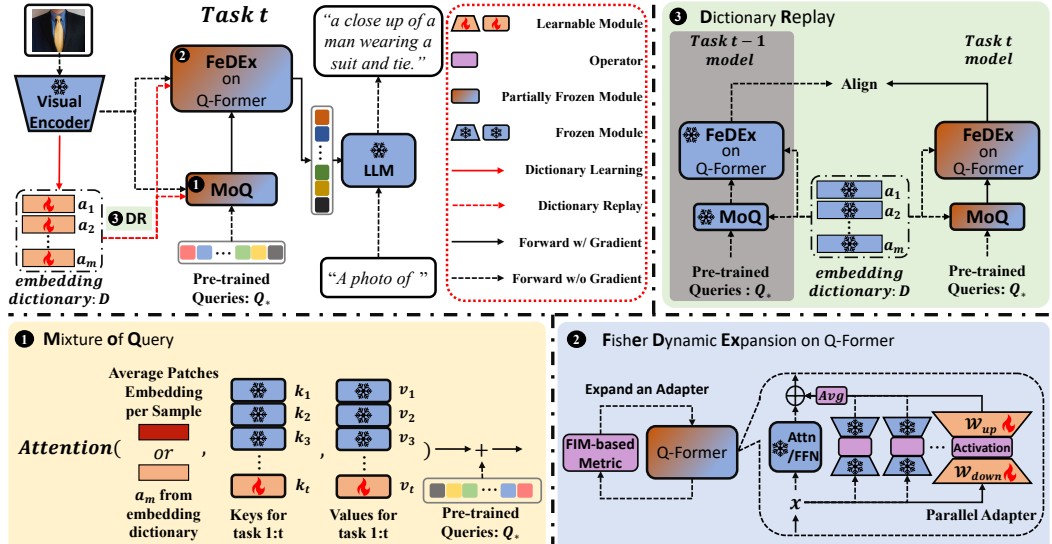

Figure 2: The framework of our exemplar-free incremental learning approach, ECA, for image-to-text generation. **Upper Left:** An input image is processed by a frozen visual encoder to produce features. These features enter the **M**ixture **o**f **Q**uery module (❶), which generates query tokens to interact with the Q-Former equipped with **F**isher **D**ynamic **Ex**pansion (❷), yielding language-informative visual representations. The representations are fed to the LLM as soft visual prompts to generate text conditioned on visual context. After the current task, visual features update the embedding dictionary via sparse dictionary learning. **Upper Right:** During training, the **D**ictionary **R**eplay module (❸) replays the embedding dictionary to retain the former alignment.

task-$t$. To determine the contributions of each $v_t$, we then introduce a task-specific key $k_t \in \mathbb{R}^{d_v}$, where $d_v$ matches the dimension of $f_{\theta_\star}(x_{t,i})$. For each sample in task-$t$, we obtain the image embeddings as $e_{t,i} = f_{\theta_\star}(x_{t,i})$, and then compute the average embedding over all the patches as $\bar{e}_{t,i}$. After that, we use an attention mechanism $Attention(\cdot, \cdot, \cdot)$ (Vaswani et al., 2017) and combine with fixed pre-trained query tokens $Q_\star$ to obtain final query tokens $Q_{t,i}$ for each image $x_{t,i}$ as below,

$$Q_{t,i} = Q_\star + Attention\left(\bar{e}_{t,i}, K_t, V_t\right), \tag{3}$$

where keys $K_t$ and task-specific query tokens $V_t$ are collected as $K_t = [k_1, k_2, \ldots, k_t] \in \mathbb{R}^{t \times d_v}$, $V_t = [v_1, v_2, \ldots, v_t] \in \mathbb{R}^{t \times n_Q \times d_Q}$ respectively.

To preserve distinct, task-specific $v_t$ and $k_t$, it is crucial that the newly learned query tokens are uncorrelated with those from previous tasks. To decrease the interference across tasks, we fix the previously learned query tokens $V_{<t}$, keys $K_{<t}$, and the pre-trained query tokens $Q_\star$, and enforce an orthogonality constraint between $(v_t, k_t)$ and $(V_{<t}, K_{<t})$ as follows,

$$\mathcal{L}_{\text{orth}}(k_t, v_t) = \|v_t V_{<t}^\top\|_F^2 + \|k_t K_{<t}^\top\|_F^2, \tag{4}$$

where $\|\cdot\|_F$ is Frobenius norm.

In addition, to ensure that each task-specific key $k_t$ is relevant to visual embeddings in task-$t$, we optimize the key-alignment objective as follows,

$$\mathcal{L}_{\text{key}}(k_t) = \frac{1}{n_t} \sum_{i=1}^{n_t} \left( 1 - \frac{k_t}{\|k_t\|_2} \frac{\bar{e}_{t,i}^\top}{\|\bar{e}_{t,i}\|_2} \right). \tag{5}$$

Finally, we optimize the MoQ as follows,

$$\mathcal{L}_{\text{MoQ}} = \mathcal{L}_{\text{orth}} + \mathcal{L}_{\text{key}}. \tag{6}$$

This attention-based aggregation strategy effectively integrates task-specific query tokens, ensuring robust continual alignment across diverse tasks.

By leveraging the MoQ module to adapt the pre-trained query tokens on a per-sample basis, the Q-Former alignment module is provided with an updated query token input, denoted as $Q_{t,i}$. Consequently, the alignment module can be reformulated as $A(\cdot; \omega_t, Q_{t,i})$.

## 4.3 FISHER DYNAMIC EXPANSION

**Motivation.** Fine-tuning the entire alignment module for each task is both costly and prone to degrading pre-trained performance. We therefore adopt a Parameter-Efficient Fine-Tuning (PEFT) approach with Parallel Adapters (PAs) (He et al., 2022). Training a single PA is efficient, but its limited capacity may fail to capture task-specific features without harming prior alignment. One naive remedy is to assign a new PA to every task. Yet in OpenITG, images often contain a dominant semantic category of visual objects along with additional semantic categories of contextual elements. This produces overlapping feature distributions that encourage positive transfer across tasks, and blindly expanding a PA for each task can break this beneficial sharing. Consequently, it motivates a principled criterion for when expansion is truly needed.

**FIM-Based Metric.** To address this challenge, we propose a Fisher Information Matrix (FIM)-based metric to guide the dynamic expansion of Parallel Adapters (PAs) across tasks. For OpenITG tasks, the Q-Former, $A_{\omega_t, Q_{t,i}}(\cdot)$, is commonly fine-tuned by minimizing the negative log-likelihood of the probability for correctly generating the label text conditioned on task-$t$ visual input $x_{t,i}$, text input $p_{t,i}$, and the label sentence $s_{t,i}$ as follows,

$$P\left(s_{t,i}^j \mid s_{t,i}^{<j}, x_{t,i}, p_{t,i}; \omega_t, Q_{t,i}\right) \triangleq g_{\phi_\star}\left(s_{t,i}^{<j}, [A(e_{t,i}; \omega_t, Q_{t,i}), p_{t,i}]\right), \tag{7}$$

$$\mathcal{L}_{\text{ce}}\left(\omega_t, Q_{t,i}\right) = -\sum_{j=1}^{|s_{t,i}|} \log P\left(s_{t,i}^j | s_{t,i}^{<j}, x_{t,i}, p_{t,i}; \omega_t, Q_{t,i}\right), \tag{8}$$

which provides a basis for subsequent analysis.

Our goal is to measure how $\omega_t$ updated by Eq. 8 on dataset $\mathcal{D}_{t+1}$ for task-$t+1$ affects performance on $\mathcal{D}_t$ for task-$t$. To this end, we first obtain the $\mathcal{L}_{\mathcal{D}_t}$ as the expectation of $\mathcal{L}_{\text{ce}}$ on dataset $\mathcal{D}_t$ by $\mathcal{L}_{\mathcal{D}_t} = \mathbb{E}_{\mathcal{D}_t}[\mathcal{L}_{\text{ce}}]$. Then, we derive a FIM-based metric $S(\omega_t)$ based on the second-order Taylor expansion with FIM approximation (see Appendix E), which reflects the overall conflict between the parameter updates for the new task and the preservation of prior knowledge.

**Definition 1.** *We define the incremental and decremental impacts of a small gradient update at $\omega_t$ on dataset $\mathcal{D}_t$ as $I_+(\omega_t) \geq 0$ and $I_-(\omega_t) \leq 0$, respectively.*

**Theorem 1.** *Based on Def. 1, the performance degradation on dataset $\mathcal{D}_t$ is denoted as $\Delta\mathcal{L}_{\mathcal{D}_t}(\Delta\omega) = I_+(\omega_t) + I_-(\omega_t)$. Define the normalized FIM-based metric as*

$$S(\omega_t) = \frac{I_+(\omega_t)}{I_+(\omega_t) + |I_-(\omega_t)|} \in [0, 1] \tag{9}$$

*Then, under a small-step update and the Fisher approximation, we have*

- *If $S(\omega_t) \leq 0.5$, training on $\mathcal{D}_{t+1}$ does not degrade the performance on $\mathcal{D}_t$ (i.e., $\Delta\mathcal{L}_{\mathcal{D}_t} \leq 0$).*

- *If $S(\omega_t) > 0.5$, the update on $\mathcal{D}_{t+1}$ will degrade the performance on $\mathcal{D}_t$.*

We provide detailed derivation and proof in Appendix E that theoretically $S(\omega_t) \leq 0.5$ guarantees non-degradation on the previous task. Moreover, Appendix E.3 reports $S(\omega_t)$ sweeps showing that 0.5 yields the best performance in our settings.

**Dynamic adapter expansion.** Finally, we employ $S(\omega_t)$ to decide whether naive reuse of the current PA on $\mathcal{D}_{t+1}$ would severely impact $\mathcal{D}_t$. When the impact metric surpasses 0.5, we expand the Q-Former with a new PA. After expanding the PA for a new task, we freeze the previously trained PAs and compute the mean of all PA outputs as the final output. FeDEx enables the Q-Former to retain the alignment of previous tasks while maintaining parameter efficiency in the training process.

## 4.4 DICTIONARY REPLAY

**Motivation**. Although the MoQ module and FeDEx integrate new task-specific knowledge while preserving previously learned knowledge, the absence of data from former tasks remains a critical challenge for exemplar-free IL. To deal with that challenge in conventional classification settings, various methods (Zhu et al., 2022; Petit et al., 2023; Zhu et al., 2021) have shown that a single prototype per category can effectively serve as a compact memory representation for data from

the previous tasks. However, as discussed in Sec. 4.2, the diverse and dispersed nature of visual embeddings in OpenITG makes a single prototype insufficient.

**Embedding Dictionary Update.** To overcome this limitation, we propose an embedding dictionary, a memory mechanism that decouples visual embeddings and captures their essential components through dictionary learning. Specifically, we learn an over-complete dictionary $D \in \mathbb{R}^{m \times d_v}$ ($m \gg d_v$) so each patch embedding $e_k \in \mathbb{R}^{d_v}$ from any image can be reconstructed as a sparse linear combination of a few rows from $D$ via solving the following Lasso problem,

$$\alpha_k = \arg\min_{\alpha \in \mathbb{R}^m} \frac{1}{2}\|e_k - D_{t-1}^\top \alpha\|_F^2 + \gamma\|\alpha\|_1, \text{ s.t. } \alpha \geq 0, \tag{10}$$

where $\gamma$ controls the sparsity of the reconstruction coefficients $\alpha_k$. The Lasso problem can be efficiently solved by FISTA (Beck & Teboulle, 2009).

After obtaining $\alpha_k$, we minimize the reconstruction error to update the embedding dictionary,

$$D_t = \arg\min_{D \in \mathbb{R}^{m \times d_v}} \frac{1}{2}\|e_k - D^\top \alpha_k\|_F^2, \text{ s.t. } \|a_j\|_2 \leq 1, \forall j \in [m], \tag{11}$$

where $a_j \in \mathbb{R}^{1 \times d_v}$ denotes $j$-th atom in $D$. The unit-norm constraint $\|a_j\|_2 \leq 1$ prevents any atom from growing too large, and removes scale ambiguity between $D$ and $\alpha_k$. With this constraint, we fix $\gamma = 1$ to obtain consistently sparse codes across tasks.

**Embedding Dictionary Replay.** With the updated embedding dictionary $D_t$ capturing the key components of visual embeddings across $t$ tasks, we replay this dictionary to retain previous task knowledge in future task training. Let $sg(\Omega_t) = sg(\{\omega_t, K_t, V_t\})$ represent the fixed updated parameters, where $sg(\cdot)$ means the stop-gradient operator. $\Omega_t$ includes updated Q-Former parameters, keys, and query tokens obtained after training task-$t$. To optimize the parameters for the future task-$t + 1$, represented by $\Omega_{t+1}$, we apply the knowledge distillation loss as follows,

$$\mathcal{L}_{\text{DR}}(\Omega_{t+1}) = \frac{1}{m}\|A(D_t; sg(\Omega_t)) - A(D_t; \Omega_{t+1})\|_F^2. \tag{12}$$

Replaying the former data encoded in the embedding dictionary ensures that the former alignment is preserved in the future task.

### 4.5 OBJECTIVE FUNCTION

Finally, to optimize the model in the task-$t$, we first use $S(\omega_t)$ in Eq. 9 to expand the Q-Former equipped with FeDEx, and then we jointly train the model parameters $\omega_t$, keys $k_t$, and values $v_t$ by the loss function with each sample in $\mathcal{D}_t$ as follows,

$$\mathcal{L} = \mathcal{L}_{\text{ce}} + \mathcal{L}_{\text{MoQ}} + \lambda\mathcal{L}_{\text{DR}}, \tag{13}$$

where $\lambda$ is a hyper-parameter that balances the contributions of the dictionary replay loss $\mathcal{L}_{\text{DR}}$. After training task-$t$, we apply the dictionary learning to update the embedding dictionary for embedding the essential elements of task-$t$'s visual embeddings.

## 5 EXPERIMENTS

To assess the proposed ECA, we first construct four new IL benchmarks for OpenITG tasks, ToS-COCO Caption, ToS-VQAv2, ToS-TextCaps, ToS-TextVQA, based on two well-known image captioning datasets, i.e. COCO ImageCaption (Lin et al., 2014), TextCaps (Sidorov et al., 2020), and two well-known VQA datasets, i.e. VQAv2 (Goyal et al., 2017), TextVQA (Singh et al., 2019). Next, we compare our proposed ECA with state-of-the-art (SOTA) exemplar-free methods on four new IL benchmarks. Lastly, we perform ablation studies to explore the impact of key components.

**Benchmarks.** To evaluate the performance of our method, we construct four IL benchmarks for OpenITG by splitting tasks based on "main topic": ToS-COCO Caption and ToS-VQAv2 from COCO Caption/VQAv2, and ToS-TextCaps and ToS-TextVQA from TextCaps/TextVQA. These splits preserve realistic overlap, where earlier main topics reappear as context in later tasks Fig. 3. The detailed introduction of constructing benchmark datasets is presented in Appendix B.

Table 1: Evaluation on ToS-COCO Caption and ToS-VQAv2. **Bold**: Best results on each dataset. Underline: Second best results on each dataset. "Avg": Final Average performance; "BWT": Backward Transfer; "FWT": Forward Transfer.

| Tasks | | ToS-COCO Caption | | | | | | | | | ToS-VQAv2 | | | |
|---|---|---|---|---|---|---|---|---|---|---|---|---|---|---|
| | | BLEU-4 | | | CIDEr | | | SPICE | | | | VQA Acc | | |
| Method | # Trainable Params | Avg ↑ | BWT ↑ | FWT ↑ | Avg ↑ | BWT ↑ | FWT ↑ | Avg ↑ | BWT ↑ | FWT ↑ | # Trainable Params | Avg ↑ | BWT ↑ | FWT ↑ |
| ZeroShot | 0 M | 36.00 | - | - | 104.65 | - | - | 21.12 | - | - | 0 M | 48.33 | - | - |
| Vanilla (PA) | 12.29 M | 42.70 | -1.49 | 6.48 | 123.00 | -4.50 | 19.15 | 23.39 | -0.78 | 2.64 | 21.74 M | 64.39 | -2.00 | 12.02 |
| Vanilla (Q-Former) | 107.13 M | 42.21 | -2.02 | 6.60 | 123.66 | -4.58 | 19.23 | 23.42 | -0.85 | 2.68 | 163.82 M | 64.14 | -1.92 | 11.78 |
| LwF (Li & Hoiem, 2017) | 12.29 M | 42.91 | -1.07 | 6.56 | 123.88 | -3.78 | 19.20 | 23.51 | -0.69 | 2.71 | 21.74 M | 64.92 | -0.99 | 14.65 |
| EWC (Lee et al., 2017) | 12.29 M | 42.86 | -1.45 | 6.62 | 123.66 | -4.03 | 18.73 | 23.55 | -0.57 | 2.55 | 21.74 M | 59.63 | -2.32 | 11.33 |
| Dual-Prompt (Wang et al., 2022e) | 14.30 M | 43.03 | **-0.62** | 6.77 | 123.59 | -1.60 | 19.04 | 23.47 | -0.52 | 2.65 | 21.67 M | 65.03 | 1.27 | 12.74 |
| CODA-Prompt (Smith et al., 2023) | 15.41 M | 43.10 | -0.67 | 6.90 | 124.20 | **-1.19** | 19.44 | 23.52 | -0.38 | 2.59 | 24.37 M | 65.64 | 1.38 | 13.71 |
| MoE-LoRA (Liu et al., 2024) | 98.84 M | 42.20 | -1.56 | 6.25 | 122.77 | -3.53 | 17.76 | 23.39 | -0.68 | 2.51 | 195.71 M | 61.02 | -3.90 | 10.27 |
| **ECA (Ours)** | 12.29 M | **43.42** | -0.64 | **7.39** | **125.56** | -1.86 | **20.58** | **23.80** | **-0.35** | **3.00** | 21.74 M | **68.05** | **1.81** | **16.38** |
| Upper-bound (PA) | 12.29 M | 43.94 | - | - | 126.91 | - | - | 24.18 | - | - | 21.74 M | 68.18 | - | - |
| Upper-bound (Q-Former) | 107.13 M | 43.82 | - | - | 126.19 | - | - | 24.10 | - | - | 163.82 M | 68.52 | - | - |

Table 2: Evaluation on ToS-TextCaps and ToS-TextVQA. **Bold**: Best results on each dataset. Underline: Second best results on each dataset. "Avg": Final Average performance; "BWT": Backward Transfer; "FWT": Forward Transfer.

| Tasks | | ToS-TextCaps | | | | | | | | | ToS-TextVQA | | |
|---|---|---|---|---|---|---|---|---|---|---|---|---|---|
| | | BLEU-4 | | | CIDEr | | | SPICE | | | VQA Acc | | |
| Method | # Trainable Params | Avg ↑ | BWT ↑ | FWT ↑ | Avg ↑ | BWT ↑ | FWT ↑ | Avg ↑ | BWT ↑ | FWT ↑ | Avg ↑ | BWT ↑ | FWT ↑ |
| ZeroShot | 0 M | 13.99 | - | - | 48.65 | - | - | 11.48 | - | - | 14.83 | - | - |
| Vanilla (PA) | 21.74 M | 24.50 | -3.14 | 9.89 | 89.39 | -2.76 | 33.61 | 15.19 | -1.47 | 3.32 | 24.94 | -3.66 | 10.85 |
| Vanilla (Q-Former) | 163.82 M | 26.98 | -1.56 | 8.66 | 93.63 | -1.84 | 28.90 | 15.90 | -0.87 | 3.29 | 32.13 | -1.37 | 15.03 |
| LwF (Li & Hoiem, 2017) | 21.74 M | 27.89 | -1.06 | 10.89 | 97.46 | 1.90 | 36.46 | 16.19 | -0.05 | 3.42 | 32.92 | -0.53 | 15.52 |
| EWC (Lee et al., 2017) | 21.74 M | 23.90 | -2.67 | 9.16 | 86.98 | -5.87 | 32.34 | 14.60 | -1.13 | 2.64 | 30.21 | 0.54 | 11.67 |
| Dual-Prompt (Wang et al., 2022e) | 19.82 M | 23.67 | -0.34 | 6.34 | 83.47 | 1.15 | 22.12 | 14.69 | **0.79** | 2.01 | 25.64 | 1.75 | 8.14 |
| CODA-Prompt (Smith et al., 2023) | 22.13 M | 24.81 | -0.48 | 7.27 | 86.33 | 1.08 | 24.70 | 15.29 | 0.77 | 2.31 | 26.13 | 1.35 | 8.83 |
| MoE-LoRA (Liu et al., 2024) | 195.71 M | 25.16 | -0.94 | 10.10 | 90.58 | -1.54 | 34.39 | 15.80 | 0.11 | 3.24 | 31.76 | -3.20 | 12.40 |
| **ECA (Ours)** | 21.74 M | **30.05** | **-0.18** | **12.13** | **103.03** | 1.94 | **39.22** | **16.86** | 0.14 | **4.39** | **38.13** | **2.36** | **19.30** |
| Upper-bound (PA) | 21.74 M | 30.59 | - | - | 110.49 | - | - | 17.78 | - | - | 41.05 | - | - |
| Upper-bound (Q-Former) | 163.82 M | 31.32 | - | - | 111.99 | - | - | 18.02 | - | - | 46.02 | - | - |

**Protocol.** Following the common protocol of the OpenITG tasks (Li et al., 2023; Del Chiaro et al., 2020; Antol et al., 2015), we use metrics BLEU@4, CIDEr, and SPICE for Image Captioning tasks, and VQA Accuracy for open-ended VQA. We assess IL performance using three metrics: *Average Performance* (Avg), *Forward Transfer* (FWT), and *Backward Transfer* (BWT), following the protocol in (Lopez-Paz & Ranzato, 2017). We report these metrics to summarize final performance across all tasks, quantify how learning later tasks affects earlier ones, and measure transfer to unseen tasks, respectively. Formal definitions and formulas are provided in Appendix C.

**Baselines.** We apply a pre-trained BLIP-2 (Li et al., 2023) as a backbone, and instantiate *all methods* on the Q-Former from the BLIP-2 with the visual encoder and the LLM frozen for fairness. Under IL settings, we evaluate "ZeroShot," "Vanilla (PA)," finetuning one Parallel Adapter (PA) on Q-Former sequentially, "Upper-bound (PA)," jointly finetuning one PA across tasks, "Vanilla (Q-Former)," finetuning the Q-Former sequentially, and "Upper-bound (Q-Former)," jointly finetuning the Q-Former across tasks. To show our performance, we also compare several state-of-the-art exemplar-free IL methods, originally developed for uni-modal tasks, namely EWC (Lee et al., 2017), LwF (Li & Hoiem, 2017), CODA-Prompt (Smith et al., 2023), Dual-Prompt (Wang et al., 2022e), and one multi-modal IL method, MoE-LoRA (Liu et al., 2024) (following (Chen et al., 2025)). LwF (Li & Hoiem, 2017) and EWC (Lee et al., 2017) are applied to the "Vanilla (PA)" under the same trainable scope as ECA for fair comparison, while other methods follow their original configuration. Full adaptation, training details, and explanations for trainable parameters are provided in Appendix D.

## 5.1 MAIN RESULTS

In this section, we compare the overall performance of our proposed method, ECA, with various baselines on our proposed benchmarks under the IL setting.

**Evaluation on ToS-COCO Caption and ToS-VQAv2.** As shown in Tab. 1, ECA significantly outperforms other baselines in terms of Avg, BWT, and FWT on both caption metrics and VQA accuracy, while using fewer trainable parameters. On ToS-COCO Caption, since BLIP-2 is pre-

Table 3: Ablation study on ToS-COCO Caption. **"Naive-Q:"** per-task query tokens without cross-task sharing (one set per task). **"DR(r):"** replay a randomly initialized dictionary (no dictionary learning).

| Method | BLEU-4 | | | CIDEr | | | SPICE | | |
|---|---|---|---|---|---|---|---|---|---|
| | Avg ↑ | BWT ↑ | FWT ↑ | Avg ↑ | BWT ↑ | FWT ↑ | Avg ↑ | BWT ↑ | FWT ↑ |
| Vanilla (PA) | 42.70 | -1.49 | 6.48 | 123.00 | -4.50 | 19.15 | 23.39 | -0.78 | 2.64 |
| PA+Naive-Q | 42.37 | -1.88 | 6.91 | 122.74 | -4.27 | 19.29 | 23.33 | -0.79 | 2.82 |
| PA+MoQ | 42.80 | -1.25 | 6.77 | 123.67 | -3.66 | 19.05 | 23.47 | -0.59 | 2.71 |
| PA+MoQ+DR | 42.97 | -1.16 | 7.28 | 124.57 | -2.80 | 20.45 | 23.59 | -0.54 | 2.96 |
| PA+MoQ+DR(r) | 42.49 | -1.57 | 7.24 | 123.24 | -3.75 | 19.88 | 23.57 | -0.66 | 2.86 |
| PA+MoQ+FeDEx | 43.22 | -0.72 | 7.05 | 124.95 | -2.04 | 19.72 | 23.69 | -0.42 | 2.83 |
| **ECA (Ours)** | **43.42** | **-0.64** | **7.39** | **125.56** | **-1.86** | **20.58** | **23.80** | **-0.35** | **3.00** |

trained on COCO Caption, the absolute gain is modest, yet the Upper-Bound Gap Closed (UBGC)[1] is remarkable. ECA's UBGC relative to CODA-Prompt, the SOTA uni-modal exemplar-free IL method, in Avg are 38.10%, 50.18%, and 42.42% for BLEU-4, CIDEr, and SPICE, respectively. Although ECA has slightly lower BWT on CIDEr than CODA-Prompt and Dual-Prompt, its higher Avg and closeness to the upper bound better reflect final IL quality. For ToS-VQAv2, whose annotations are not part of BLIP-2's pre-training, ECA is only 0.13 below the **upper-bound** performance and surpasses CODA-Prompt by 2.41 in terms of Avg for VQA accuracy.

**Evaluation on ToS-TextCaps and ToS-TextVQA** These two benchmarks form a harder continual-alignment setting because BLIP-2 was not pre-trained on them, and successful predictions rely on OCR tokens that must interact with visual features through cross-attention. As shown in Tab. 2, ECA remains strong on both datasets. On ToS-TextCaps, compared with LwF as the best uni-modal exemplar-free baseline and with the "Upper-bound (PA)" which uses the same trainable scope under joint training, ECA improves Avg by 2.16, 5.57, and 0.67 for BLEU-4, CIDEr, and SPICE, with UBGC of 80.00%, 42.74%, and 42.14%. On ToS-TextVQA, ECA improves Avg VQA accuracy over LwF by 5.21, with a UBGC of 64.08%.

**Findings** We further examine these exemplar-free methods and observe several notable points: **(1).** As shown in Tab. 1 and Tab. 2, LwF surpasses prompt based baselines on ToS-TextCaps and ToS-TextVQA. These datasets require reasoning over OCR tokens, which induces a larger distribution shift beyond the pretraining regime. Prompt pools do not explicitly preserve the alignment for the newly introduced tokens, whereas LwF maintains the cross-modal alignment through knowledge distillation. **(2).** EWC is misaligned with our main topic setting. Classical EWC presumes disjoint tasks and thus restricts updates to parameters deemed important for past tasks. In our scenario, cross-task semantic overlap means such updates can be beneficial. Enforcing these constraints suppresses useful sharing and harms transfer. As shown in Tab. 2, EWC even underperforms the "Vanilla (PA)." **(3).** As shown in Tab. 1 and Tab. 2, ECA achieves impressive BWT and FWT across all metrics on four datasets. This indicates that ECA not only mitigates catastrophic forgetting but also uses previously learned knowledge to better handle future tasks. Namely, ECA attains a broad understanding of tasks with different main topics and generalizes well to upcoming tasks. **(4).** As shown in Tab. 1 and Tab. 2, ECA uses almost the same number of trainable parameters as the baselines with single PA, yet consistently outperforms them and even methods with much larger parameter budgets. This suggests that ECA better exploits a limited alignment capacity rather than blindly increasing model size and computational cost. Furthermore, we show its parameter and inference efficiency in Appendix H.

In sum, ECA delivers superior overall performance and achieves continual alignment for IL in OpenITG with efficient parameter usage. We also show additional case studies in Appendix J.

## 5.2 ABLATION STUDIES

**Effect of key components & Hyper-parameters.** We first study the effect of key components of ECA in Tab. 3, MoQ improves Avg and BWT over "Vanilla (PA)" by sharing orthogonal query tokens across tasks, DR further boosts, especially FWT via dictionary replay, and FeDEx mitigates forgetting by expanding adapters only when needed. The detailed analyses are in Appendix F. Additional ablation for losses in MoQ shown in Tab. 9, Appendix F. Moreover, we also study the influence of hyper-parameters, i.e. DR weight $\lambda$, DR's embedding dictionary atom number $m$ in Appendix G.

---

[1]**UBGC** $= (\text{ECA} - \text{method})/(\text{Upper-bound (PA)} - \text{method}) \times 100\%$; UB is the oracle joint-training upper bound on the union of tasks.

## 6 LIMITATIONS AND FUTURE WORK

While ECA effectively mitigates catastrophic forgetting under our main-topic IL setting for OpenITG, it has several natural limitations that suggest directions for future work. First, Dictionary Replay (DR) maintains a single embedding dictionary across tasks in an online setting. However, as task sequences are increased with highly diverse visual distributions, a predefined size of the dictionary may not be large enough. Atoms that are heavily used by the current task may be reused and updated by later tasks with very different visual distributions, potentially overwriting earlier representations and causing extra forgetting. Thus, extending our current fixed-size dictionary to a dictionary of dynamic, adaptive size is an interesting avenue for future research. Second, our current instantiation of ECA assumes a reasonably strong pre-trained VLM, where the visual encoder and language model already provide high-quality representations. In scenarios with weaker backbones or limited pre-training data, solely using ECA may be insufficient. Extending ECA to weaker VLMs or jointly coupling pre-training and continual alignment is left as future work.

## 7 CONCLUSION

In this work, we introduced the notion of continual alignment for incremental learning in open-ended image-to-text generation. Based on this idea, we proposed Efficient Continual Alignment (ECA), an exemplar-free framework for the alignment module in VLMs. ECA employed three key components: the Mixture of Query (MoQ) module, the Fisher Dynamic Expansion (FeDEx), and Dictionary Replay (DR). These components allowed for pre-trained VLM to acquire new, task-specific features while preserving robust cross-modal alignment without storing exemplars. We also constructed four new IL benchmarks that simulate realistic distribution shifts in OpenITG. Extensive experiments demonstrated that ECA could significantly mitigate catastrophic forgetting and achieve superior performance with high parameter efficiency.

ETHICS STATEMENT

Our work studies exemplar-free incremental learning (IL) for open-ended image-to-text generation (OpenITG) by adapting only the alignment module of pre-trained VLMs, BLIP-2. We evaluate on publicly available datasets MSCOCO (Caption), VQAv2, TextCaps, and TextVQA in Sec. 5 without collecting new human data or processing personal identifiers. Our work does not involve private or sensitive information. To build the topic-split benchmarks, we rely on COCO annotations or GPT-4o object detection prompts, and we release our topic-split benchmarks on an anonymized GitHub repository at abstract.

REPRODUCIBILITY STATEMENT

We are committed to reproducible research. All datasets are public, and our benchmarks are released in the anonymized GitHub repository linked in the abstract. We will release the full code and scripts upon publication. Model architectures, training settings, and hyper-parameters are specified in the paper and Appendix D.

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

## LLM USAGE

We used GPT-4o to assign main topics to images when constructing the ToS benchmarks and to verify label consistency, as presented in Appendix B. We also used an LLM for light grammar editing.

## APPENDIX

## APPENDIX CONTENTS

## A  NOTATION LIST.

Table 4: Summary of Notations.

| Notation | Definition |
|---|---|
| $t$ | The current task ID |
| $\mathcal{D}_t = \{(X_t, P_t, S_t)\}$ | The dataset for task-$t$ |
| $X_t = \{x_{t,i}\}$ | The set of input images for task-$t$ |
| $P_t = \{p_{t,i}\}$ | The set of associated text inputs (prompts or questions) for task-$t$ |
| $S_t = \{s_{t,i}\}$ | The set of ground-truth sentences for each image-text pair |
| $n_t$ | The number of instances in task-$t$ |
| $\hat{s}_{t,i}$ | The predicted sentence for image $x_{t,i}$ |
| $\theta_t$ | Trainable parameters of the visual encoder (if fine-tuned) |
| $\theta_\star$ | Frozen parameters of the pre-trained visual encoder |
| $\phi_t$ | Trainable parameters of the Large Language Model (LLM) at task-$t$ (if fine-tuned) |
| $\phi_\star$ | Frozen parameters of the pre-trained Large Language Model (LLM) |
| $f_\theta(\cdot)$ | Visual encoder with parameters $\theta$ |
| $A_{\omega_t, Q_t}(\cdot)$ | Alignment module (Q-Former) with trainable parameters $\omega_t$ and learnable queries $Q_t$ |
| $g_\phi(\cdot)$ | Large Language Model (LLM) with parameters $\phi$ |
| $\omega_t$ | Trainable parameters of the Q-Former, including attention layers and a fully connected layer for query projection |
| $d_v$ | Dimension of the visual embeddings |
| $Q_t$ | Learnable query tokens |
| $n_Q$ | The number of learnable query tokens in Q-Former |
| $d_Q$ | The dimension of each query token in Q-Former |
| $v_t$ | Task-specific query tokens in the Mixture of Query (MoQ) module for task-$t$ |
| $k_t$ | Task-specific key in the Mixture of Query (MoQ) module for task-$t$ |
| $e_{t,i} = f_{\theta_\star}(x_{t,i})$ | Image embeddings |
| $\bar{e}_{t,i}$ | Average embedding over all patches in $e_{t,i}$ |
| $e_k$ | The $k$-th compact visual embedding in the dictionary |
| $K_t = [k_1, k_2, \ldots, k_t]$ | Collected keys for all tasks |
| $V_t = [v_1, v_2, \ldots, v_t]$ | Collected query tokens for all tasks |
| $S(\omega_t)$ | FIM-based metric for measuring conflict in parameter updates |
| $I_+(\omega_t)$ | Incremental contribution to $\Delta\mathcal{L}_{\mathcal{D}_t}$ |
| $I_-(\omega_t)$ | Decremental contribution to $\Delta\mathcal{L}_{\mathcal{D}_t}$ |
| $\Omega_t = \{\omega_t, K_t, V_t\}$ | Learnable Parameters for task-$t$ |
| $D \in \mathbb{R}^{m \times d_v}$ | Overcomplete dictionary of compressed visual embeddings |
| $\alpha_k$ | Sparse coding coefficients |
| $\|\cdot\|_F$ | Frobenius norm |
| $m$ | Number of dictionary atoms in the embedding dictionary |
| $\gamma$ | Regularization weight in sparse coding |

Table 4 describes notations used in our work.

## B  NEW BENCHMARK

To evaluate the proposed method under a more realistic scenario, we propose a new setting for IL in OpenITG tasks. Unlike previous benchmarks (Del Chiaro et al., 2020; Qian et al., 2023) that group images into five disjoint categories and remove images with objects from multiple categories, as we described at Sec. 1, we split the tasks based on the image's "main topic". We define an image's main topic as the semantic category of its most prominent object. In our scenario, grouping images by main topic reflects real-world conditions, where a single image may predominantly contain one semantic

category while also including other contextual elements. Thus, as shown in Fig. 3, the distribution of semantic categories changes as the main topic evolves. This design effectively simulates the dynamic distribution shifts caused by environmental and temporal changes.

Based on this setting, we construct four new IL benchmarks for OpenITG. For the COCO Caption and VQAv2 tasks based on the COCO ImageCaption (Lin et al., 2014) dataset, we label our benchmarks as ToS-COCO Caption and ToS-VQAv2. We first extract the area, class, and super-category of each foreground object from MSCOCO instance labels. The super-category of the object with the largest area is assigned as the image's main topic. Initially, 12 super-categories are obtained: People, Animal, Vehicle, Outdoor, Sports, Kitchen, Food, Furniture, Electronic, Appliance, Indoor, and Accessory. However, some main topics, such as "People" and "Kitchen" frequently appear across multiple main topics and are considered "common topics." Images labeled with these common topics are reassigned based on the next largest non-common object. The final benchmark comprises 10 main topics: "Animal", "Vehicle", "Outdoor", "Sports", "Food", "Furniture", "Electronics", "Appliances", "Indoor", "Accessories." For images without instance information, we employ GPT-4o with specific instructions to detect prominent foreground objects and assign classes and main topics accordingly.

For the TextCaps and TextVQA tasks, we follow a similar procedure and label our benchmarks as ToS-TextCaps and ToS-TextVQA. Since TextCaps provides only object classes and may include inaccuracies, we use GPT-4o with specific prompts to verify and detect the prominent foreground objects. Following the 10 main topics defined in MSCOCO, we assign each image a main topic. In our experiments, we observe that very few images have "Animal" as their main topic, so the final benchmark comprises 9 main topics, consistent with MSCOCO except for "Animal."

For ToS-COCO Caption, we adopt the train, validation, and test splits from prior work (Karpathy & Fei-Fei, 2015) on COCO Caption and then partition the data into 10 tasks based on our labeled main topics. For ToS-VQAv2, ToS-TextCaps, and ToS-TextVQA, we use the official train-validation splits of VQAv2, TextCaps, and TextVQA and further divide them into 10, 9, and 9 tasks, respectively, according to our main topic labels.

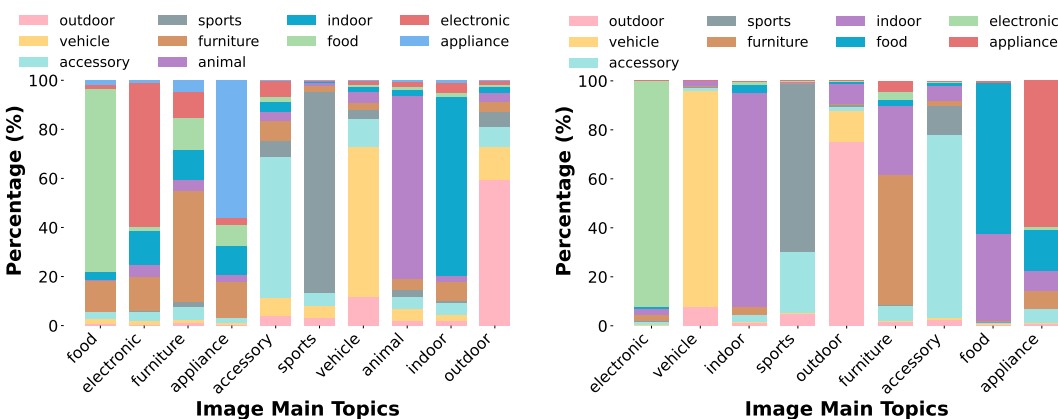

Figure 3: The distribution of topics in four benchmarks. **Left:** ToS-COCO Caption & ToS-VQAv2. **Right:** ToS-TextCaps & ToS-TextVQA.

## C    EXPERIMENTS PROTOCOL

Following the common protocol of the OpenITG tasks (Li et al., 2023; Del Chiaro et al., 2020; Antol et al., 2015), we use metrics BLEU@4, CIDEr, and SPICE for Image Captioning tasks, and VQA Accuracy for open-ended VQA. We assess IL performance using three metrics: *Average Performance* (Avg), *Forward Transfer* (FWT), and *Backward Transfer* (BWT), following the protocol in (Lopez-Paz & Ranzato, 2017). Let $r_{t,\tau}$ denote the OpenITG metric value of task-$\tau$ after training on task-$t$, $\bar{b}_\tau$ the OpenITG metric value for task-$\tau$ on the initial model parameters, and $T$ the total task number. Then, we can obtain the Avg of the final task as $Avg = \frac{1}{T}\sum_{i=1}^{T} r_{T,i}$, the BWT as $BWT = \frac{1}{T-1}\sum_{i=1}^{T-1} r_{T,i} - r_{i,i}$, and the FWT as $FWT = \frac{1}{T-1}\sum_{i=2}^{T} r_{i-1,i} - \bar{b}_i$. We report these

metrics to capture the model's overall performance on all tasks, the effect of learning new tasks on past performance, and the model's ability to transfer knowledge to unseen tasks, respectively.

## D  EXPERIMENTS DETAIL

**Training Details.**  In our experiments, we use the pre-trained BLIP-2 (Li et al., 2023), which includes pre-trained ViT-g/14 from EVA-CLIP (Fang et al., 2023) as the frozen visual encoder, an unsupervised-trained OPT-2.7B (Zhang et al., 2022a) as the frozen LLM, and a pre-trained Q-Former. Then we instantiate ECA and all other methods at the Q-Former from the pre-trained BLIP-2, and we compare their performance across different datasets. For all approaches, we use the same optimizer hyper-parameters as in the original BLIP-2.

To apply CODA-Prompt and Dual-Prompt, we follow the original works (Smith et al., 2023; Wang et al., 2022e) and insert deep prompts into the self-attention layers of the Q-Former. Specifically, for CODA-Prompt, prompts are inserted into layers 1–5 of the Q-Former, matching the depth of both the Q-Former and ViT-B/16 (Dosovitskiy et al., 2020). The prompt length is set to 8. Based on the ablation studies in the original papers, we increase the prompt pool size to 30 prompts per task (300 in total for 10 tasks) for ToS-COCO Caption and 50 prompts per task (500 in total for 10 tasks) for the rest of the datasets. This adjustment is made to ensure parameter efficiency, fairness, and to enhance performance. The orthogonal loss weight is set to 0.1 following the original configuration. For Dual-Prompt, we insert G-prompts into layers 1–2 and E-prompts into layers 3–5. Similarly, we increase both G-prompt and E-prompt length to 500 for ToS-COCO Caption and 800 for the rest of the datasets. The balance factor in Dual prompt is set to 1 following the original configuration.

For our ECA, within the Q-Former, FeDEx inserts Parallel Adapters (PAs) in every self-attention and feed-forward layer. The low-rank for PAs is set to 30 for attention layers and 512 for feed-forward layers, with all PA scales set to 4 according to (He et al., 2022). For computational efficiency, we replace the non-linear activation with an identity function so that multiple frozen PAs can be merged during training and inference. For the embedding dictionary, we set the number of atoms as $m = 5 \times d_v$, where $d_v$ matches the dimension of the frozen visual encoder $f(x_{t,i}; \theta_\star)$. The weight of DR loss, $\lambda$ in Eq. 13, is set to 0.1, and the threshold of $S(\omega_t)$ in Eq. 9 is set to 0.5 for all experiments.

For MoE-LoRA (Chen et al., 2025), we follow the original paper by setting 8 experts per layer and inserting MoE-LoRA into every feed-forward network, with each expert having a low-rank of 512, consistent with our ECA configuration.

For LwF and EWC (Lee et al., 2017), we directly apply them into the backbone, "Vanilla (PA)," to report the performance for fair comparison.

During training, we extract image features as in BLIP-2 and provide additional tokens to the Q-Former. Specifically, following Q-Former inputs, we input the question token for ToS-VQAv2, the official OCR token for ToS-TextCaps, and both OCR and question tokens for ToS-TextVQA to enable interaction with query tokens via the self-attention layers. These tokens introduce a small number of trainable parameters in the embedding layer and feed-forward network inside the Q-Former, so the reported number of trainable parameters differs across datasets. Training hyper-parameters are detailed in Tab. 5.

Table 5: Hyper-parameters for fine-tuning ECA

| **Datasets** | **ToS-COCO Caption** | **ToS-VQAv2** | **ToS-TextCaps** | **ToS-TextVQA** |
|---|---|---|---|---|
| Fine-tuning epochs | 5 | 5 | 5 | 5 |
| Warm-up steps per task | 100 | 100 | 100 | 100 |
| Learning rate | 1e-05 | 1e-5 | 1e-5 | 1e-5 |
| Batch size | 64 | 64 | 32 | 32 |
| AdamW $\beta$ | (0.9, 0.999) | (0.9, 0.999) | (0.9, 0.999) | (0.9, 0.999) |
| Weight decay | 0.05 | 0.05 | 0.05 | 0.05 |
| Drop path | 0 | 0 | 0 | 0 |
| Image resolution | 364 | 490 | 364 | 490 |
| Inference beam size | 5 | 5 | 5 | 5 |
| Prompt | a photo of | Question: {} Short answer: | Based on OCR: {}. A photo of | Based on OCR: {}. Question: {} Short answer: |
| Atom Number $m$ | 7040 | 7040 | 7040 | 7040 |
| $\gamma$ | 1 | 1 | 1 | 1 |

# E   DERIVATION AND PROOF OF THEOREM 1

Here we provide the details of the derivation and proof of Theorem 1.

**Theorem 1.** *Based on Def. 1, the performance degradation on dataset $\mathcal{D}_t$ is denoted as $\Delta\mathcal{L}_{\mathcal{D}_t}(\Delta\omega) = I_+(\omega_t) + I_-(\omega_t)$. Define the normalized FIM-based metric as*

$$S(\omega_t) = \frac{I_+(\omega_t)}{I_+(\omega_t) + |I_-(\omega_t)|} \in [0, 1] \tag{9}$$

*Then, under a small-step update and the Fisher approximation, we have*

- *If $S(\omega_t) \leq 0.5$, training on $\mathcal{D}_{t+1}$ does not degrade the performance on $\mathcal{D}_t$ (i.e., $\Delta\mathcal{L}_{\mathcal{D}_t} \leq 0$).*

- *If $S(\omega_t) > 0.5$, the update on $\mathcal{D}_{t+1}$ will degrade the performance on $\mathcal{D}_t$.*

## E.1   DERIVATION OF THEOREM 1

First, we introduce how to obtain the incremental and decremental contributions for $\mathcal{L}_{\mathcal{D}_t}(\omega_t)$ Our goal is to measure how $\omega_t$, updated by Eq. 8 on dataset $\mathcal{D}_{t+1}$ for task-$t+1$, affects the performance on $\mathcal{D}_t$ for task-$t$. To this end, we first obtain the loss on $\mathcal{D}_t$ as the expectation of the cross-entropy loss $\mathcal{L}_{ce}$ on dataset $\mathcal{D}_t$:

$$\mathcal{L}_{\mathcal{D}_t} = \mathbb{E}_{\mathcal{D}_t}[\mathcal{L}_{ce}].$$

Then, we perform a second-order Taylor expansion of the loss $\mathcal{L}_{\mathcal{D}_t}(\omega)$ around $\omega_t$:

$$\Delta\mathcal{L}_{\mathcal{D}_t}(\Delta\omega) \triangleq \mathcal{L}_{\mathcal{D}_t}(\omega_t + \Delta\omega) - \mathcal{L}_{\mathcal{D}_t}(\omega_t)$$

$$\approx \sum_{i=1}^{N} \nabla_{\omega_i}\mathcal{L}_{\mathcal{D}_t}(\omega_t)\,\Delta\omega_i$$

$$+ \frac{1}{2}\sum_{i=1}^{N} H_{\mathcal{D}_t}^i(\omega_t)\,(\Delta\omega_i)^2, \tag{14}$$

where $N$ is the number of parameters in $\omega_t$, and $H_{\mathcal{D}_t}^i(\omega_t)$ denotes the $i$-th diagonal element of the Hessian of $\mathcal{L}_{\mathcal{D}_t}$.

In theory, the full Hessian $H_{\mathcal{D}_t}(\omega_t)$ contains off-diagonal elements that capture parameter interactions. However, computing the full Hessian is computationally prohibitive, and in many practical scenarios the off-diagonal elements are relatively small compared to the diagonal. Therefore, it is common to approximate the Hessian by its diagonal. Moreover, under the negative log-likelihood loss (i.e., for $\mathcal{L}_{\mathcal{D}_t}(\cdot)$), the diagonal of the Hessian is well approximated by the diagonal of the Fisher Information Matrix (FIM) (Kirkpatrick et al., 2017). Hence, we set

$$H_{\mathcal{D}_t}^i(\omega_t) \approx F_{\mathcal{D}_t}^i(\omega_t).$$

Since we use the (empirical) Fisher approximation, $F_{\mathcal{D}_t}^i(\omega_t) \geq 0$ for all $i$.

Under a small-step gradient descent update on $\mathcal{D}_{t+1}$ with 1 learning rate, a typical update for each parameter is given by

$$\Delta\omega_i = -\nabla_{\omega_i}\mathcal{L}_{\mathcal{D}_{t+1}}(\omega_t).$$

Substituting this into Eq. 14 yields the per-parameter impact function:

$$I(\omega_{t,i}) = -\nabla_{\omega_i}\mathcal{L}_{\mathcal{D}_t}(\omega_t)\,\nabla_{\omega_i}\mathcal{L}_{\mathcal{D}_{t+1}}(\omega_t)$$

$$+ \frac{1}{2}F_{\mathcal{D}_t}^i(\omega_t)\left(\nabla_{\omega_i}\mathcal{L}_{\mathcal{D}_{t+1}}(\omega_t)\right)^2. \tag{15}$$

A large positive $I(\omega_{t,i})$ indicates that updating $\omega_{t,i}$ on $\mathcal{D}_{t+1}$ would significantly impair the alignment on $\mathcal{D}_t$.

Furthermore, to better capture the overall impact on the model parameters, we aggregate these per-parameter impacts by distinguishing the positive and negative contributions:

$$I_+(\omega_t) = \sum_{i=1}^{N} \max\{0,\, I(\omega_{t,i})\}, \tag{16}$$

$$I_-(\omega_t) = \sum_{i=1}^{N} \min\{0, I(\omega_{t,i})\}. \tag{17}$$

Then we define the normalized conflict metric as

$$S(\omega_t) = \frac{I_+(\omega_t)}{I_+(\omega_t) + |I_-(\omega_t)|} \in [0, 1]. \tag{18}$$

The metric $S(\omega_t)$ reflects the overall conflict between parameter updates for the new task and the preservation of prior knowledge.

### E.2 PROOF OF THEOREM 1

*Proof.* By definition, we have:

$$\Delta\mathcal{L}_{\mathcal{D}_t}(\Delta\omega) = I_+(\omega_t) + I_-(\omega_t).$$

Since $I_+(\omega_t) \geq 0$ and $I_-(\omega_t) \leq 0$, we set $|I_-(\omega_t)| = -I_-(\omega_t)$ and rewrite the conflict metric as:

$$S(\omega_t) = \frac{I_+(\omega_t)}{I_+(\omega_t) + |I_-(\omega_t)|}.$$

Then, $\Delta\mathcal{L}_{\mathcal{D}_t}(\Delta\omega) \leq 0$ holds if and only if:

$$I_+(\omega_t) \leq |I_-(\omega_t)|,$$

which, after dividing by the positive term $(I_+(\omega_t) + |I_-(\omega_t)|)$, gives:

$$S(\omega_t) \leq 0.5.$$

Conversely, if $S(\omega_t) \leq 0.5$, then:

$$I_+(\omega_t) \leq |I_-(\omega_t)| \implies I_+(\omega_t) + I_-(\omega_t) \leq 0.$$

Similarly, if $S(\omega_t) > 0.5$, then $I_+(\omega_t) > |I_-(\omega_t)|$, which implies:

$$\Delta\mathcal{L}_{\mathcal{D}_t}(\Delta\omega) = I_+(\omega_t) + I_-(\omega_t) > 0.$$

Thus, the equivalences hold in both directions, completing the proof. $\square$

### E.3 EXPERIMENT ON $S(\omega)$

To verify our Theorem 1, we sweep the threshold of $S(\omega_t)$ and observe that $S(\omega_t) = 0.5$ consistently attains the best trade-off across architectures and metrics (Fig. 4). Thresholds substantially larger than $0.5$ tend to reduce performance due to unnecessary adapter expansion.

## F EFFECT OF KEY COMPONENTS.

We examine how each component influences model performance. (1) As shown in Tab. 6, adding the MoQ module to "Vanilla (PA)" raises the Avg and improves BWT, indicating reduced catastrophic forgetting. (2) For "PA+MoQ" versus "PA+Naive-Q", learning separate task-specific query tokens for each task reduces Avg and makes BWT more negative, while FWT becomes slightly higher. This pattern reflects topic overlap. MoQ addresses it through shared and orthogonalized query tokens across tasks. Comparing "PA+MoQ+DR" with "PA+MoQ," and "PA+MoQ+FeDEx" with "ECA," reveals that DR substantially boosts IL performance, especially FWT. This occurs because the embedding dictionary, built via sparse dictionary learning, decouples visual embeddings into essential components, so replaying them benefits future tasks. Lastly, the differences between "PA+MoQ+FeDEx" and "PA+MoQ," as well as between "ECA" and "PA+MoQ+DR," show that FeDEx significantly increases performance and mitigates forgetting by selectively expanding Parallel Adapters based on Eq. 9, allowing new knowledge to be integrated without disrupting previously

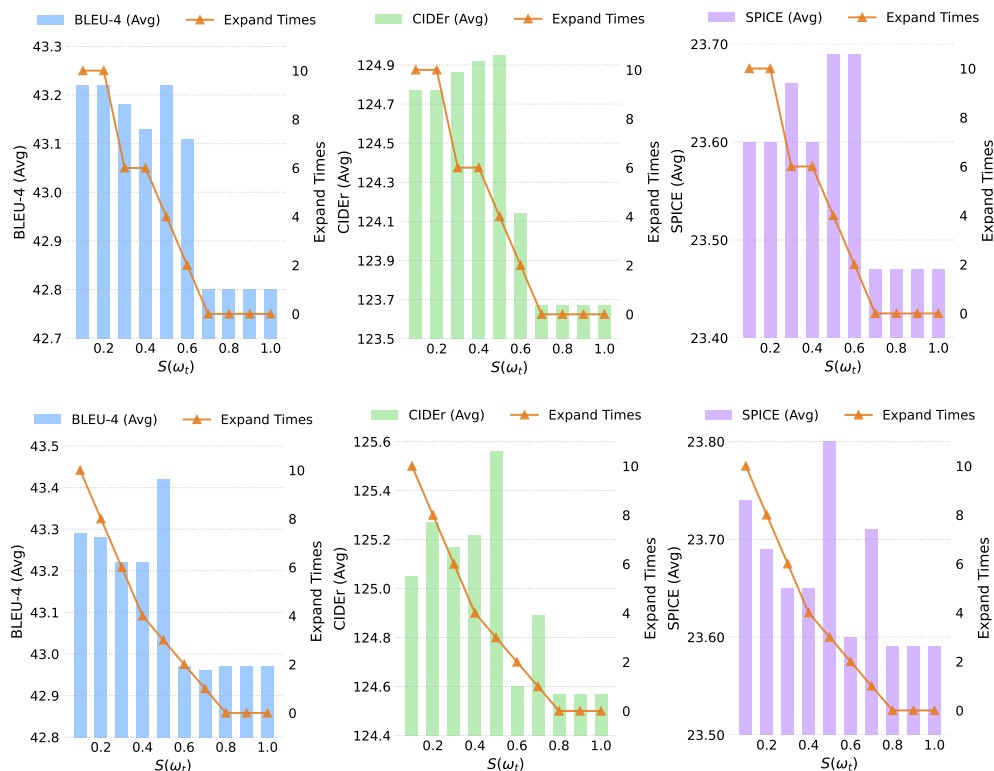

Figure 4: Performance of different structures with different threshold of $S(\omega_t)$ in FeDEx. **Upper Line:** the performance of "PA+MoQ+FeDEx"; **Bottom Line:** the performance of the whole "ECA."

Table 6: Ablation study on ToS-COCO Caption. **"Naive-Q:"** per-task query tokens without cross-task sharing (one set per task). **"DR(r):"** replay a randomly initialized dictionary (no dictionary learning).

| Method | BLEU-4 | | | CIDEr | | | SPICE | | |
|---|---|---|---|---|---|---|---|---|---|
| | Avg ↑ | BWT ↑ | FWT ↑ | Avg ↑ | BWT ↑ | FWT ↑ | Avg ↑ | BWT ↑ | FWT ↑ |
| Vanilla (PA) | 42.70 | -1.49 | 6.48 | 123.00 | -4.50 | 19.15 | 23.39 | -0.78 | 2.64 |
| PA+Naive-Q | 42.37 | -1.88 | 6.91 | 122.74 | -4.27 | 19.29 | 23.33 | -0.79 | 2.82 |
| PA+MoQ | 42.80 | -1.25 | 6.77 | 123.67 | -3.66 | 19.05 | 23.47 | -0.59 | 2.71 |
| PA+MoQ+DR | 42.97 | -1.16 | 7.28 | 124.57 | -2.80 | 20.45 | 23.59 | -0.54 | 2.96 |
| PA+MoQ+DR(r) | 42.49 | -1.57 | 7.24 | 123.24 | -3.75 | 19.88 | 23.57 | -0.66 | 2.86 |
| PA+MoQ+FeDEx | 43.22 | -0.72 | 7.05 | 124.95 | -2.04 | 19.72 | 23.69 | -0.42 | 2.83 |
| **ECA (Ours)** | **43.42** | **-0.64** | **7.39** | **125.56** | **-1.86** | **20.58** | **23.80** | **-0.35** | **3.00** |

learned alignments. Additionally, comparing "PA+MoQ+DR" and "PA+MoQ+DR(r)" shows that leveraging dictionary learning to decouple visual embeddings into essential components is important.

To show the importance of each loss in MoQ, i.e. $\mathcal{L}_{\text{orth}}$ and $\mathcal{L}_{\text{key}}$ in Eq. 6, we compare their performance separately. As shown in Tab. 9, the $\mathcal{L}_{\text{orth}}$ can significantly increase the BWT, which means it decreases the influence of the previously learned query tokens. Regarding the $\mathcal{L}_{\text{key}}$, optimizing only by $\mathcal{L}_{\text{key}}$ without the $\mathcal{L}_{\text{orth}}$ may increase the interference across tasks. However, while optimizing both losses, $\mathcal{L}_{\text{key}}$ can ensure that each task-specific key is relevant to visual embeddings in task-$t$, and leverage the $\mathcal{L}_{\text{orth}}$ to preserve distinct sets of query tokens.

## G  EFFECT OF HYPER-PARAMETERS.

Then we apply the grid search to explore the effect of three hyper-parameters, i.e. DR weight $\lambda$, DR's embedding dictionary atom number $m$, and the threshold of FIM-based metric value, $S(\omega_t)$, in FeDEx. For exploring the hyper-parameters of DR, we tested a range of $\lambda$ and $m$ settings on the

Table 7: Ablation study on ToS-COCO Caption.

| Method | BLEU-4 | | | CIDEr | | | SPICE | | |
|---|---|---|---|---|---|---|---|---|---|
| | Avg ↑ | BWT ↑ | FWT ↑ | Avg ↑ | BWT ↑ | FWT ↑ | Avg ↑ | BWT ↑ | FWT ↑ |
| Vanilla (PA) | 42.70 | -1.49 | 6.48 | 123.00 | -4.50 | 19.15 | 23.39 | -0.78 | 2.64 |
| PA+MoQ ($\mathcal{L}_{key}$) | 42.76 | -1.80 | 6.66 | 123.29 | -4.34 | 18.50 | 23.37 | -0.79 | 2.71 |
| PA+MoQ ($\mathcal{L}_{orth}$) | 42.67 | -1.30 | 6.92 | 123.21 | -3.99 | 19.84 | 23.40 | -0.68 | 2.88 |
| PA+MoQ ($\mathcal{L}_{orth} + \mathcal{L}_{key}$) | 42.80 | -1.25 | 6.77 | 123.67 | -3.66 | 19.05 | 23.47 | -0.59 | 2.71 |

(a) "PA+MoQ+DR" with different $\lambda$

| $\lambda$ | 0.01 | 0.05 | 0.1 | 0.5 | 1 |
|---|---|---|---|---|---|
| BLEU-4 (Avg) | 42.61 | 42.34 | **42.97** | 42.71 | 42.39 |
| CIDEr (Avg) | 123.38 | 122.64 | **124.57** | 123.01 | 122.84 |
| SPICE (Avg) | 23.54 | 23.29 | **23.59** | 23.59 | 23.49 |

(b) "PA+MoQ+DR" with different $m = M \times d_v$

| $m$ | 2.5x | 5x | 7.5x | 10x | 12.5x |
|---|---|---|---|---|---|
| BLEU-4 (Avg) | 41.63 | **42.97** | 42.65 | 42.56 | 42.85 |
| CIDEr (Avg) | 122.42 | **124.57** | 123.22 | 123.38 | 123.59 |
| SPICE (Avg) | 23.53 | **23.59** | 23.54 | 23.46 | 23.48 |

Table 8: Ablations of Hyper-parameters in DR on ToS COCO Caption

"PA+MoQ+DR" structure. As shown in Tab. 8a and Tab. 8b, we set $\lambda = 0.1$ in all experiments and $m = 5 \times d_v$.

## H  PARAMETER AND INFERENCE EFFICIENCY

Table 9: Parameter and inference efficiency analyze on ToS-COCO Caption.

| Metric | Method | | | | | | | |
|---|---|---|---|---|---|---|---|---|
| | Vanilla (PA) | Vanilla (Q-Former) | LwF | EWC | Dual-Prompt | CODA-Prompt | MoE-LoRA | **ECA (Ours)** |
| Trainable Params ↓ | 12.29M | 107.13M | 12.29M | 12.29M | 14.30M | 15.41M | 98.84M | 12.29M |
| Training GPU Memory ↓ | 18.80G | 21.50G | 18.80G | 18.80G | 19.47G | 19.55G | 21.37G | 18.92G |
| Inference GPU Memory ↓ | 10.67G | 10.65G | 10.67G | 10.67G | 10.70G | 10.70G | 11.02G | 10.72G |
| Throughput (token/s) ↑ | 36.52 | 37.62 | 36.52 | 36.52 | 32.82 | 34.77 | 36.37 | 36.49 |

In this section, we further analyze the parameter and inference efficiency of methods on an NVIDIA A40 GPU. As shown in Tab. 9, we compare parameter and runtime efficiency on the ToS-COCO Caption benchmark. All methods share the same pre-trained BLIP-2 backbone and are evaluated on the same GPU with an identical batch size and input length.

Comparing the Trainable Params and training/inference GPU memory usage in Tab. 9, ECA uses almost the same number of trainable parameters and peak GPU memory as baselines with single PA (i.e. Vanilla (PA), LwF, EWC), while matching or even slightly exceeding their inference throughput. In contrast, methods with much larger parameter budgets (i.e. Dual-Prompt, CODA-Prompt, MoE-LoRA) are slower and require more memory. These results support our claim that ECA enhances continuous performance with parameter efficiency and that it better leverages a limited alignment capacity rather than blindly expanding the model.

## I  ECA ON PROJECTOR-BASED MULTI-MODAL LLMS

In this section, we further illustrate how to instantiate our proposed ECA on a projector-based multi-modal LLM (MLLM), e.g. LLaVA Liu et al. (2023). Unlike the Q-Former in BLIP-2, which uses learnable query tokens to directly interact with visual embeddings, LLaVA adopts a visual *projector* that maps visual features into the language token space, and then relies on self-attention in the LLM to achieve visual–language alignment. In this case, we regard the top $L$ layers of the LLM as the effective alignment module. Based on this view, Fig. 5 illustrates how we instantiate ECA on a projector-based MLLM.

As shown in Fig. 5, a frozen visual encoder followed by a pre-trained projector produces visual tokens. These visual tokens enter the Mixture-of-Query (MoQ) module, which learns task-specific soft prompts and attentively aggregates them. The visual tokens, aggregated soft prompts, and textual prompts are concatenated and fed into the LLM, whose top $L$ layers are equipped with FeDEx.

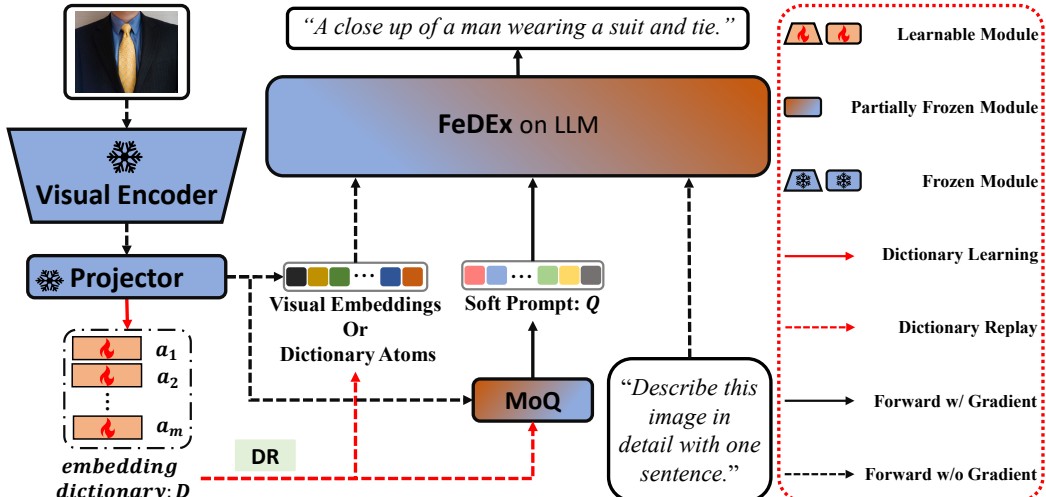

Figure 5: The framework of our ECA instantiated on a projector-based multi-modal LLM (e.g. LlaVA). An input image is processed by a frozen visual encoder with a pre-trained projector to produce visual tokens. These visual tokens enter the **M**ixture **of Q**uery module, which generates soft prompt token and catted with visual tokens to interact with the LLM equipped with **F**isher **D**ynamic **Ex**pansion, to generate text conditioned on visual context. After the current task, visual tokens update the embedding dictionary via sparse dictionary learning. During training, the **D**ictionary **R**eplay module replays the embedding dictionary to retain the former alignment.

FeDEx selectively expands parallel adapters in these layers based on the FIM-based conflict score, so that new features are incorporated while preserving established alignment. Meanwhile, Dictionary Replay (DR) maintains an embedding dictionary and replays it during training to retain information from previous tasks.

For DR on projector-based MLLMs, we use the dictionary atoms as visual tokens and pass them, together with soft prompts generated fro MoQ, and a textual prompt, into the LLM. Concretely, we first use the model trained by previous tasks (teacher) to generate pseudo captions conditioned on the dictionary atoms, soft prompts and textual prompt. We then feed the same inputs into the current training model (student) and compute a token-level KL divergence between the teacher and student predictive distributions as a knowledge distillation loss. This allows the embedding dictionary to replay past visual semantics without storing raw images, encouraging the projector-based MLLM with ECA to preserve alignment learned from earlier tasks.

## J CASE STUDY

In this section, we show the comparison on real cases between our ECA and the CODA-Prompt, which is the second-best uni-modal exemplar-free baseline on the ToS-VQAv2. As shown in Fig. 6, we use real cases of the first main topic after training BLIP-2 by both methods on all tasks from ToS-VQAv2 to test the model. The CODA-Prompt ruined the alignment established from previous tasks while training the model sequentially. However, our ECA aligns both modalities continually, and finally, it can still respond in great shape to the previously learned knowledge. The reason is that the CODA-Prompt only uses soft prompts to learn the new knowledge, which is limited. Thus, it is hard to achieve continual alignment for VLM only by the prompt when the data has complex semantics, and tasks have overlapping semantics. We also compare cases from the middle task, which is closer to the final task. As shown in Fig. 7, the CODA-Prompt still can not understand well on some easy cases, while our ECA still shows a great performance on mitigating catastrophic forgetting.

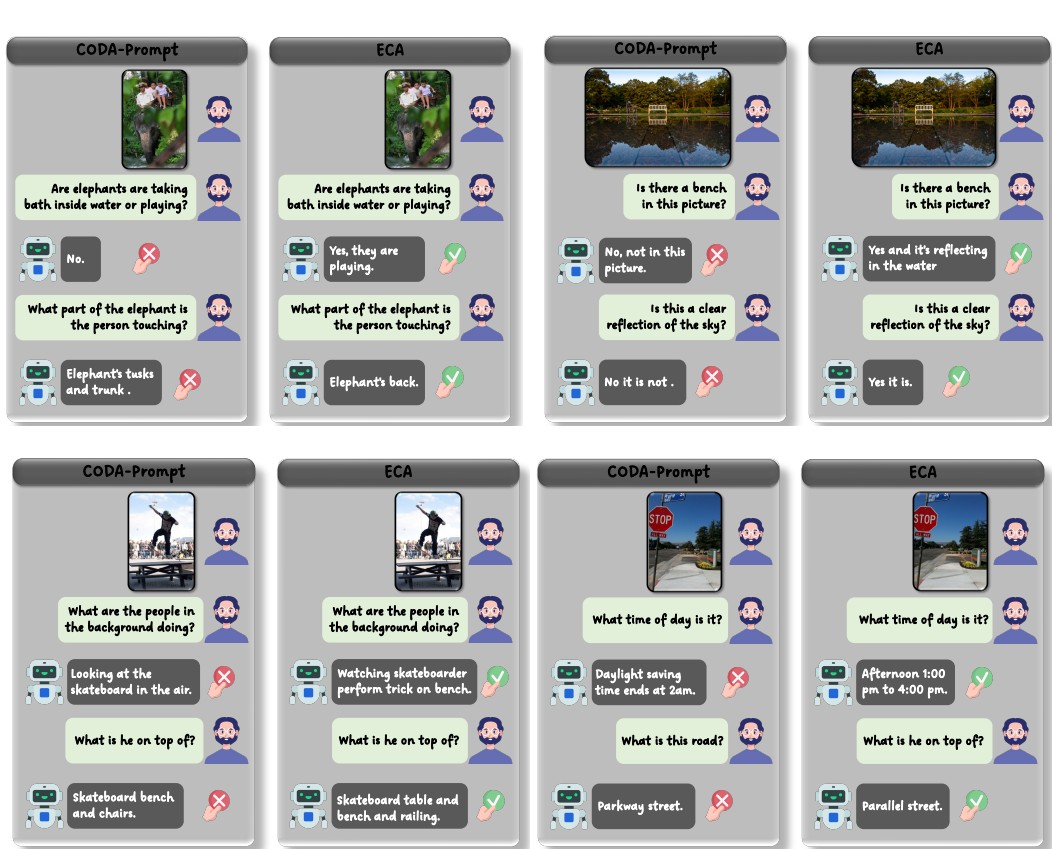

Figure 6: Comparison on the case from the first task in ToS-VQAv2.

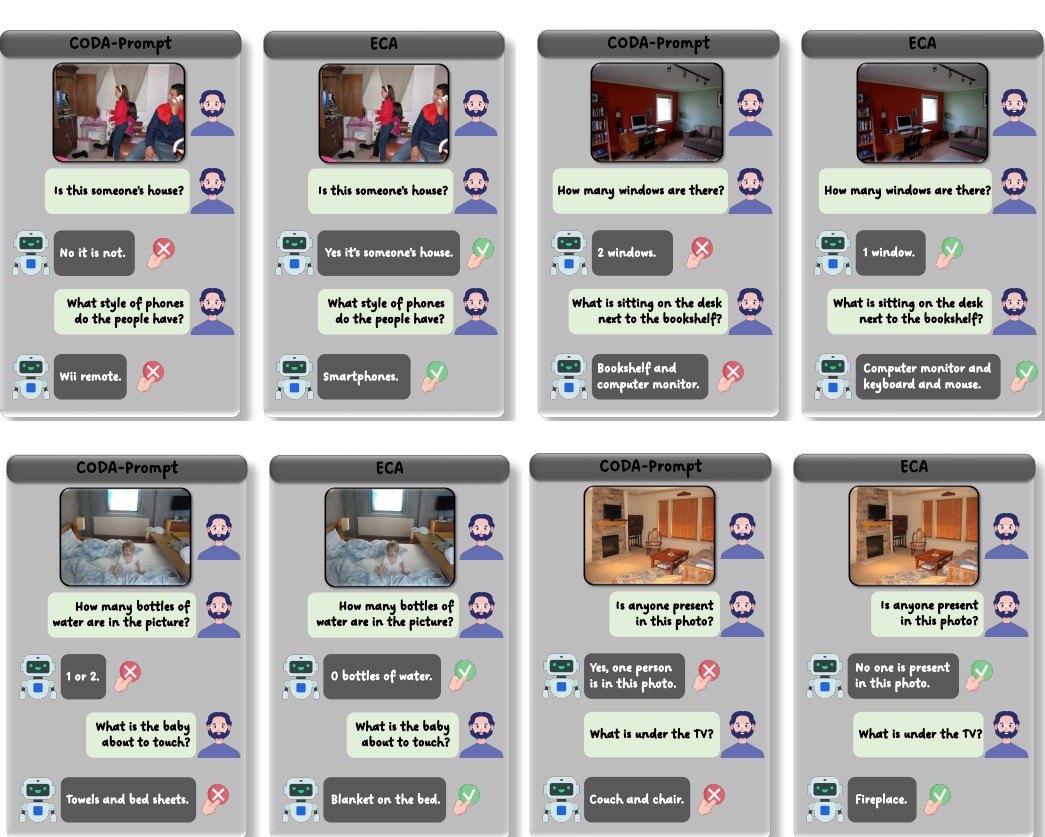

Figure 7: Comparison on the case from the middle task in ToS-VQAv2.

