# OpenReview forum: "ECA: Efficient Continual Alignment for Open-Ended Image-to-Text Generation"
_ICLR.cc/2026/Conference — Submitted to ICLR 2026_

### Official Review · Reviewer_phKC · 2025-10-28

**Soundness:** 4
**Presentation:** 4
**Contribution:** 3
**Rating:** 4
**Confidence:** 3

**Summary:**

This paper addresses the problem of Incremental Learning (IL) for Open-ended Image-to-Text Generation (OpenITG), such as VQA and captioning.

Experiments on the four new benchmarks show that ECA significantly outperforms strong baselines (including regularization and prompt-based IL methods) and achieves performance very close to the joint-training upper bound, demonstrating its ability to mitigate catastrophic forgetting while remaining parameter-efficient.

**Strengths:**

1. The "main topic" shift scenario, which incorporates semantic overlap, is a significant and more realistic formulation for IL benchmarks compared to standard disjoint-category setups. The four new ToS benchmarks are a strong contribution.
2. The FIM-based metric for deciding when to expand with a new parallel adapter is the paper's strongest technical novelty. It is theoretically motivated (Theorem 1) and empirically validated (Fig. 4), providing a non-heuristic way to balance positive transfer (reusing adapters) and mitigating interference (adding new adapters).
3. The combination of MoQ and DR (Dictionary Replay) is a highly effective *exemplar-free* approach. The DR mechanism, using a learned sparse dictionary of embeddings, is a clever way to perform rehearsal without storing raw data, addressing privacy and storage concerns.

**Weaknesses:**

1. The final ECA method combines three distinct modules (MoQ, FeDEx, DR), each with its own logic and (minor) hyper-parameters (e.g., $\lambda$ for $\mathcal{L}_{DR}$, dictionary size $m$). This is inherently more complex than a simpler baseline like a single PA or a prompt-based method.
2. The paper does not explicitly quantify the computational cost of the FeDEx module. Calculating the FIM-based metric $S(\omega_t)$ requires additional gradient and FIM diagonal computations at the end of each task to decide whether to expand. While likely manageable (as it's not per-batch), a brief analysis of this overhead would be beneficial.
3. The method is instantiated on BLIP-2's Q-Former. While the *principles* (adapting the alignment module) are general, it's not immediately clear how MoQ or FeDEx would be applied to different VLM architectures, such as projector-based models (e.g., LLaVA), which lack a Q-Former. A brief discussion on this potential for generalization would improve the paper.

**Questions:**

1. Could you please quantify the computational overhead of the FeDEx module? Specifically, what is the cost of calculating the FIM-based metric $S(\omega_t)$ at the end of each task (relative to the task's training time)?
2. The Dictionary Replay (DR) module relies on a dictionary of a fixed size $m=5 \times d_v$. How does this fixed-size dictionary scale as the number of tasks $T$ grows very large? Do you foresee this becoming a bottleneck, and would a dynamic-sized dictionary (e.g., adding new atoms per task) be beneficial?
3. Could you elaborate on how the core ideas of ECA, particularly FeDEx and MoQ, could be adapted to other popular VLM architectures that do not use a Q-Former, such as those with simple MLP projectors?

---

> ### Author Response · Authors · 2025-11-21
>
> Thank you for your constructive comments and suggestions! We have addressed all the comments and suggestions from the Reviewer and accordingly updated our manuscript highlighted in BLUE. We hope our responses below address your concerns. Please let us know if you have any additional concerns.
>
> Q1: Please quantify the computational overhead of computing the FIM-based metric $S(\omega)$ at each task end relative to the task’s training time.
> -
> **A1:** Per your suggestion, we quantify the computational cost of FeDEx. In our experiments, the overhead of FeDEx is less than $20\%$ of the per-task training time, since a full training pass includes two forward–backward passes, while calculating the FIM-based metric only requires one pass. We explain the details below.
>
> As shown in Appendix E.1, we calculate the $S(\omega)$ by $I(\omega)$, where $I(\omega\_{t,i}) =-\nabla\_{\omega\_i} \mathcal{L}\_{\mathcal{D}\_t}(\omega\_t)\nabla\_{\omega\_i} \mathcal{L}\_{\mathcal{D}\_{t+1}}(\omega\_t)+\frac{1}{2}F^i\_{\mathcal{D}\_t}(\omega\_t)\Bigl(\nabla\_{\omega\_i} \mathcal{L}\_{\mathcal{D}\_{t+1}}(\omega\_t)\Bigr)^2$, and $\mathcal{L}\_{\mathcal{D}\_t} = \mathbb{E}\_{\mathcal{D}\_t}[\mathcal{L}\_{ce}]$. The diagonal Fisher term $F^i\_{\mathcal{D}\_t}$ is computed in the standard way as $\mathbb{E}\_{\mathcal{D}\_t}[(\nabla\_{\omega\_i} \mathcal{L}\_{ce}(\omega\_t))^2]$. Thus, we only need to run the cross-entropy loss on the current dataset $\mathcal{D}\_t$ and the next task's dataset $\mathcal{D}\_{t+1}$ to obtain the gradients, square them, and accumulate the averages. In addition to optimizing the cross-entropy loss, a normal training step needs to further use the dictionary atoms to calculate the dictionary replay loss. Thus, a full training pass includes two forward and backward passes, while calculating the FIM-based metric only requires one pass. Therefore, each FIM pass is cheaper than a full training pass.
>
> Since each task is trained for 5 epochs in our experiments, two CE-only passes per task boundary correspond to at most about one additional epoch of computation. Therefore, the overhead of FeDEx is less than $20\%$ of the per-task training time. In practice, we can also estimate $S(\omega)$ on a subset of each dataset for further efficiency, which makes the actual wall-clock overhead even smaller.
>
> Q2: How does the fixed-size DR dictionary scale as the number of tasks grows, and could a dynamic dictionary (e.g., adding atoms per task) help avoid potential bottlenecks?
> -
> **A2:** We agree that any fixed-capacity replay mechanism involves a trade-off: for very long task sequences with highly diverse visual distributions, a single fixed dictionary may eventually become a capacity bottleneck. We explain and discuss the details as below.
>
> In DR, we mainly have two reasons to design as a fixed-size, **over-complete dictionary** $D \in \mathbb{R}^{m \times d_v}$, where $m = 5 d_v$ and $m\gg d_v$, in our experiments, which is shared across tasks. **(i) memory efficiency**: this choice keeps the memory footprint of DR at $O(m d_v)$, independent of the number of tasks $T$. In contrast, per-task dictionaries would grow roughly linearly with $T$. **(ii) balancing feature sharing and interference**: The sparse representations brought by the overcomplete dictionary can effectively decrease the harmful interference between tasks. On the other hand, the shared dictionary can learn basic visual features shared across tasks for better maintaining previous tasks' knowledge to mitigate catastrophic forgetting. However, if we expand the dictionary for each task, the dictionary size will be increased linearly, and it is hard to reuse previously learned basic features to learn the current knowledge efficiently.
> This design follows the classical sparse dictionary learning paradigm, where a fixed over-complete dictionary plus sparse coding is used to represent diverse visual features via sparse combinations of shared atoms [1,2].
>
> For very long task sequences with highly diverse visual distributions, the predefined size of the dictionary is not large enough. However, we can extend our current fixed-size dictionary to a dictionary of dynamic, adaptive size: specifically, if the harmful interference between tasks exceeds a certain threshold, we add more atoms to the dictionary. **We discuss this future work in our "Limitations and Future Work" (Sec. 6 in the revised version)**.
>
> [1] Sparse coding with an overcomplete basis set: a strategy employed by V1? (Vision Research 1997)
>
> [2] K‑SVD: An Algorithm for Designing Overcomplete Dictionaries for Sparse Representation (IEEE Transactions on Signal Processing 2006)

---

> ### Author Response · Authors · 2025-11-21
>
> Q3: How can ECA be adapted to VLMs without a Q-Former, such as models with simple MLP projectors?
> -
> **A3:** Per your suggestion, we have applied the proposed ECA to other projector-based MLLMs, i.e., LLaVA-v0, **as shown in Appendix I of the revised version**.
>
>
> Due to the different architectures of Q-Former-based VLMs (e.g., BLIP-2) and projector-based MLLMs (e.g., LLaVA), we extend our proposed ECA to projector-based MLLMs, as described in Appendix I.
> We then adapt Vanilla (PA), MoE-LoRA, and ECA to LLaVA-v0 and evaluate them on ToS-TextCaps, since ToS-TextCaps is not included in the published pre-training recipe of LLaVA-v0.
> This setting allows us to assess the continual learning behaviour of different methods when the model is exposed to a new captioning dataset that it has not been pre-trained on.
>
>
> As shown in Table 1, our proposed ECA achieves better performance than MoE-LoRA on LLaVA-v0 with comparable or fewer trainable parameters. Compared with Vanilla (PA), ECA also improves BWT, indicating that it can still achieve continual alignment and mitigate catastrophic forgetting on projector-based MLLMs.
>
> Table 1: Evaluation on ToS-TextCaps. All methods are adapted to LLaVA-v0.
>
> |Method|Trainable Params (M)|BLEU-4 Avg ↑|BLEU-4 BWT ↑|BLEU-4 FWT ↑|
> |-|-|-|-|-|
> |Vanilla (PA)|31.51|18.40|-2.34|6.35|
> |MoE-LoRA|208.05|23.31|-2.12|9.20|
> |**ECA (ours)**|**31.51**|**26.12**|**0.61**|**10.16**|
>
> Q4: The final ECA method combines three distinct modules (MoQ, FeDEx, DR), which seems more complex than a single PA or prompt-based method.
> ------------------
>
> **A4:** We appreciate this concern. **While ECA consists of three components, they are designed as a single, coherent "continual alignment" framework.**
>
> Specifically, as we stated in Sec. 1 (L.78-88):
> 1. MoQ learns task-specific query tokens to acquire new cues with minimal disruption to prior alignment.
> 2. FeDEx selectively expands parallel adapters, preserving established alignment while allocating capacity to new topics.
> 3. DR leverages a shared dictionary to capture task-agnostic visual components across tasks.
>
> Together, these components aim to preserve established alignment while allocating capacity to new topics in OpenITG IL, where the main visual topics evolve and overlap across tasks.
>
> As shown in Tab. 3 (Sec. 5.2), where we start from a single PA baseline and gradually add MoQ, FeDEx, and DR. The ablations show consistent incremental gains from each module, while keeping the trainable-parameter count identical to the single-PA baselines and the computational overhead small (see the new efficiency table in Appendix H).

---

> ### Author Response · Authors · 2025-11-26
> **Reminder to reviewer: Looking Forward to Your Further Comments and Feedback**
>
> Dear Reviewer phKC,
>
> We would like to thank you for taking the time to review our paper and for the insightful comments. We have addressed all the comments and suggestions you made. In particular, as suggested, we have **quantified the computational overhead of FeDEx, analyzed the scalability of the fixed-size DR dictionary and discussed a potential dynamic extension, and demonstrated the applicability of ECA to projector-based VLMs such as LLaVA-v0.** As we are approaching the midpoint of the discussion period, please kindly let us know if you have any additional concerns. We truly appreciate this opportunity to improve our work and shall be most grateful for any feedback you could give to us. If you do not have further questions, we are curious if you could consider raising our score.
>
> Thank you very much!

---

### Official Review · Reviewer_ZE8G · 2025-10-30

**Soundness:** 2
**Presentation:** 4
**Contribution:** 2
**Rating:** 6
**Confidence:** 3

**Summary:**

This work proposes Efficient Continual Alignment (ECA), an exemplar-free incremental learning framework for open-ended image-to-text generation (OpenITG) that adapts pre-trained vision-language models to evolving visual domains. ECA introduces continual alignment, ensuring cross-modal consistency while learning new tasks without accessing prior data. It achieves this through three key components: a Mixture of Query (MoQ) module for task-specific query adaptation, a Fisher Dynamic Expansion (FeDEx) mechanism that expands model capacity using FIM-based metrics, and a Dictionary Replay (DR) strategy to preserve past knowledge. Together, these techniques effectively mitigate catastrophic forgetting and enhance continual generation performance in dynamic visual environments.

**Strengths:**

1. This paper presents the proposed setting and methodology through clear introductions and illustrations, along with detailed definitions.
2. This paper proposes a novel IL Benchmarks for OpenITG framework that addresses the issue of semantic overlap in image categories or background scenes across different tasks.

**Weaknesses:**

1. Given the continual changes in visual semantic themes, why only fine-tune the alignment module? In real-world scenarios, can this solution still perform well when encountering scenes or categories that the visual extractor has never seen before?
2. This paper employs BLIP-2 for experimentation. Has consideration been given to validating the method's effectiveness on more novel models? In particular, consider other forms of multimodal alignment such as Linear Projector/MLP.
3. How effective is the Fisher metrics screening in Fisher Dynamic Expansion? The lack of experimental demonstration shows how much unnecessary expansion Fisher metrics reduce during dynamic expansion. Also, the increased inference costs resulting from dynamic expansion have not been taken into account.

**Questions:**

See weaknesses.

---

> ### Author Response · Authors · 2025-11-21
>
> Thank you for your constructive comments and suggestions! We have addressed all the comments and suggestions from the Reviewer and accordingly updated our manuscript highlighted in BLUE. We hope our responses below address your concerns. Please let us know if you have any additional concerns.
>
> Q1: Why fine-tune only the alignment module, and can this approach still handle truly unseen scenes or categories for the visual encoder?
> -
> **A1:** As we stated in Sec. 1, we assume the pre-trained VLM is already a generalist model that can produce high-quality features. Thus, fine-tuning the alignment module suffices to capture the shift in visual semantic themes and achieve *continual alignment*. In addition, full-scale fine-tuning for a well-pretrained VLM is inefficient and can erode pre-training gains[1]. Therefore, we only focus on fine-tuning the alignment module for efficiency.
>
> As shown in Tab. 2 (Sec. 5), on ToS-TextCaps and ToS-TextVQA, which induce a larger distribution shift beyond the pre-training regime (L.454–455), our proposed ECA still outperforms all baselines. This verifies our assumption, i.e., if the visual extractor is well pre-trained, our proposed ECA can still perform well. **We also add a "Limitations and Future Work" section (see Sec. 6) to discuss this limitation of ECA**
>
> [1] Investigating the catastrophic forgetting in multimodal large language model fine-tuning. (CPAL 2023).
>
> Q2: Consider other forms of multimodal alignment, such as Linear Projector/MLP.
> -
> **A2:** Thanks for the suggestion. **In the revised version, we add a “ECA on projector-based Multi-modal LLMs” section as Appendix I.**
>
> Due to the different architectures of Q-Former-based VLMs (e.g., BLIP-2) and projector-based MLLMs (e.g., LLaVA), we extend our proposed ECA to projector-based MLLMs, as described in Appendix I.
> We then adapt Vanilla (PA), MoE-LoRA, and ECA to LLaVA-v0 and evaluate them on ToS-TextCaps, since ToS-TextCaps is not included in the published pre-training recipe of LLaVA-v0.
> This setting allows us to assess the continual learning behaviour of different methods when the model is exposed to a new captioning dataset that it has not been pre-trained on.
>
> As shown in Table 1, our proposed ECA achieves better performance than MoE-LoRA on LLaVA-v0 with comparable or fewer trainable parameters. Compared with Vanilla (PA), ECA also improves BWT, indicating that it can still achieve continual alignment and mitigate catastrophic forgetting on projector-based MLLMs.
>
> Table 1: Evaluation on ToS-TextCaps. All methods are adapted to LLaVA-v0.
>
> |Method|Trainable Params (M)|BLEU-4 Avg ↑|BLEU-4 BWT ↑|BLEU-4 FWT ↑|
> |-|-|-|-|-|
> |Vanilla (PA)|31.51|18.40|-2.34|6.35|
> |MoE-LoRA|208.05|23.31|-2.12|9.20|
> |**ECA (ours)**|**31.51**|**26.12**|**0.61**|**10.16**|
>
> Q3: How effective is the Fisher-based screening at reducing unnecessary expansions, and what impact does dynamic expansion have on inference cost?
> -
> **A3:** (i) We have illustrated the number of expansions and effectiveness of Fisher Dynamic Expansion under different thresholds $S(\omega)$ in Fig.4 in Appendix E.3.
>
> Specifically, as shown in Fig.4, when we set $S(\omega)=0$ (i.e., naively expanding a new PA after every task), the PA is expanded 10 times on ToS-COCO Caption. In contrast, using the theoretical threshold $S(\omega)=0.5$ (which is also adopted in all our main experiments), the PA is expanded only 3 times, while achieving better BWT/Avg performance by reusing previously learned knowledge. This shows that the Fisher-based metric is effective at avoiding unnecessary expansions while preserving the benefits of capacity increase.
>
> (ii) Regarding inference cost, according to the original Appendix D (L.942–944), "for computational efficiency we replace the non-linear activation in each PA with an identity function so that multiple frozen PAs can be merged during training and inference," expanding PAs will not increase the inference cost.
>
> Furthermore, we add a short section in **Appendix H** to discuss the efficiency, where we compare trainable parameters, peak training/inference GPU memory, and inference throughput (tokens/s) on ToS-COCO Caption under the same backbone (BLIP-2 ViT-g/14 + OPT-2.7B) and GPU (NVIDIA A40), as summarized in Table 2. Comparing the inference GPU memory usage and Inference Throughput between Vanilla (PA) and ECA, we can observe that the FeDEx in ECA will not increase the inference cost.
>
> Table 2: Comparison of parameter, runtime, and memory efficiency on ToS-COCO Caption (BLIP-2 backbone: ViT-g/14 + OPT-2.7B, on NVIDIA A40).
>
> |Method|Trainable Params (M)|Peak GPU Mem (Train, GB)| GPU Mem (Infer, GB)| Inference Throughput (tokens/s) ↑|
> |-|-|-|-|-|
> |Vanilla (PA)|12.29|18.80|10.67|36.52|
> |Vanilla (Q-Former)|107.13|21.50|10.65|37.62|
> |LwF|12.29|18.80|10.67|36.52|
> |EWC|12.29|18.80|10.67|36.52|
> |Dual-Prompt|14.30|19.47|10.70|32.82|
> |CODA-Prompt|15.41|19.55|10.70|34.77|
> |MoE-LoRA|98.84|21.37|11.02|36.37|
> |**ECA (ours)**|12.29|18.92|10.72|36.49|

---

> ### Author Response · Authors · 2025-11-26
> **Reminder to reviewer: Looking Forward to Your Further Comments and Feedback**
>
> Dear Reviewer ZE8G,
>
> We would like to thank you for taking the time to review our paper and for the insightful comments. We have addressed all the comments and suggestions you made. In particular, as suggested, we have **clarified and empirically supported why we only fine-tune the alignment module, discussed its future directions, extended ECA to a projector-based VLM (LLaVA-v0), and quantified the effect and cost of the Fisher Dynamic Expansion.** As we are approaching the midpoint of the discussion period, please kindly let us know if you have any additional concerns. We truly appreciate this opportunity to improve our work and shall be most grateful for any feedback you could give to us. If you do not have further questions, we are curious if you could consider raising our score.
>
> Thank you very much!

---

### Official Review · Reviewer_DSk8 · 2025-10-30

**Soundness:** 3
**Presentation:** 3
**Contribution:** 3
**Rating:** 6
**Confidence:** 1

**Summary:**

This paper introduces Efficient Continual Alignment (ECA), a novel exemplar-free incremental learning approach for open-ended image-to-text generation that addresses the practical scenario where visual data categories shift over time. ECA enables vision-language models to continuously adapt to new images while preserving previously learned knowledge through three core mechanisms: a Mixture of Query module for task-specific adaptation, Fisher Dynamic Expansion for strategic model growth based on Fisher Information Matrix metrics, and Dictionary Replay using an embedding dictionary to maintain past knowledge without storing raw historical data. The authors construct four new benchmarks reflecting real-world conditions and demonstrate that ECA significantly reduces catastrophic forgetting and outperforms baseline methods in incremental learning scenarios where the alignment module must continuously evolve without access to previous task data.

**Strengths:**

1. This paper addresses a more realistic scenario where visual data distributions shift over time as environments evolve, unlike previous static assumptions.
2. This paper introduces an exemplar-free approach that preserves knowledge without storing raw historical data, making it more practical and privacy-preserving.
3. This paper proposes a novel continual alignment framework with dynamic model expansion that efficiently adapts to new tasks while minimizing interference with established cross-modal representations.

**Weaknesses:**

1. Based on Figure 2, the proposed FeDEx and Q-Former appear to be the same component, which contradicts the paper's claimed contributions.
2. The manuscript does not include a limitations section. The reviewer requests that the authors provide a comprehensive discussion of the method's limitations in their rebuttal response.
3. The experimental comparisons are limited to earlier image captioning approaches and lack benchmarking against recent large-scale vision-language models (e.g., Qwen-VL and LLaVA)

**Questions:**

Please refer to above weaknesses.

---

> ### Author Response · Authors · 2025-11-21
>
> Thank you for your constructive comments and suggestions! We have addressed all the comments and suggestions from the Reviewer and accordingly updated our manuscript highlighted in BLUE. We hope our responses below address your concerns. Please let us know if you have any additional concerns.
>
> Q1: Based on Figure 2, FeDEx appears indistinguishable from the Q-Former.
> -
> **A1:** For clarity, in Fig.2, the Q-Former denotes the alignment module, and our proposed FeDEx is integrated on top of this module. Accordingly, we changed the label from “FeDEx (Q-Former)” to “FeDEx on Q-Former”. **We have updated Fig.2 in the revised version.**
>
> Q2: The manuscript does not include a limitations section.
> -
> **A2:** Thanks for the suggestion. **We have added a "Limitations and Future Work" section (Section 6 in our revised version) to discuss the limitations of ECA, summarized as follows.**
>
> First, our Dictionary Replay module learns a single embedding dictionary across tasks in an online setting. Atoms that are heavily used by the current task may be reused and updated by later tasks with very different visual distributions, potentially overwriting earlier representations and causing extra forgetting. Second, our current instantiation of ECA assumes a reasonably strong pre-trained VLM backbone that can produce high-quality representations. In settings where the backbone is poorly pre-trained or has limited generalization ability, continual alignment alone may not fully prevent catastrophic forgetting.
>
> Q3: Comparisons lack baselines from recent large VLMs (e.g., Qwen-VL, LLaVA).
> -
> **A3:** Per your suggestion, we have applied the proposed ECA to another projector-based MLLM, i.e., LLaVA-v0, **as shown in Appendix I of the revised version**.
>
> Due to the different architectures of Q-Former-based VLMs (e.g., BLIP-2) and projector-based MLLMs (e.g., LLaVA), we extend our proposed ECA to projector-based MLLMs, as described in Appendix I.
> We then adapt Vanilla (PA), MoE-LoRA, and ECA to LLaVA-v0 and evaluate them on ToS-TextCaps, since ToS-TextCaps is not included in the published pre-training recipe of LLaVA-v0.
> This setting allows us to assess the continual learning behaviour of different methods when the model is exposed to a new captioning dataset that it has not been pre-trained on.
>
> As shown in Table 1, our proposed ECA achieves better performance than MoE-LoRA on LLaVA-v0 with comparable or fewer trainable parameters. Compared with Vanilla (PA), ECA also improves BWT, indicating that it can still achieve continual alignment and mitigate catastrophic forgetting on projector-based MLLMs.
>
> Table 1: Evaluation on ToS-TextCaps. All methods are adapted to LLaVA-v0.
>
> |Method|Trainable Params (M)|BLEU-4 Avg ↑|BLEU-4 BWT ↑|BLEU-4 FWT ↑|
> |-|-|-|-|-|
> |Vanilla (PA)|31.51|18.40|-2.34|6.35|
> |MoE-LoRA|208.05|23.31|-2.12|9.20|
> |**ECA (ours)**|**31.51**|**26.12**|**0.61**|**10.16**|

---

> ### Author Response · Authors · 2025-11-26
> **Reminder to reviewer: Looking Forward to Your Further Comments and Feedback**
>
> Dear Reviewer DSk8,
>
> We would like to thank you for taking the time to review our paper and for the insightful comments. We have addressed all the comments and suggestions you made. In particular, as suggested, we have **clarified the role of FeDEx on top of the Q-Former in Fig. 2, added a "Limitations and Future Work" section, and extended our experiments to LLaVA-v0 to include a recent large VLM baseline.** As we are approaching the midpoint of the discussion period, please kindly let us know if you have any additional concerns. We truly appreciate this opportunity to improve our work and shall be most grateful for any feedback you could give to us. If you do not have further questions, we are curious if you could consider raising our score.
>
> Thank you very much!

---

### Official Review · Reviewer_sepo · 2025-10-31

**Soundness:** 3
**Presentation:** 3
**Contribution:** 3
**Rating:** 4
**Confidence:** 4

**Summary:**

This paper proposes Efficient Continual Alignment (ECA), an exemplar-free incremental learning (IL) framework for open-ended image-to-text generation (OpenITG), which addresses catastrophic forgetting by adapting only the alignment module of pre-trained vision-language models (VLMs) (e.g., BLIP-2’s Q-Former) while freezing visual encoders and large language models (LLMs). ECA integrates three key components (Mixture of Query, Fisher Dynamic Expansion, Dictionary Replay) to preserve cross-modal alignment, and the authors construct four realistic IL benchmarks (ToS-COCO Caption, ToS-VQAv2, etc.) split by image main topics, with experiments showing ECA outperforms SOTA exemplar-free baselines in average performance, forward/backward transfer.

**Strengths:**

A key strength is the introduction of "continual alignment" as a novel concept for multi-modal IL, marking the first work to explicitly target preserving the cross-modal alignment of VLM alignment modules in exemplar-free OpenITG—filling a gap in existing works that either rely on raw exemplars or ignore alignment stability.

ECA’s component design is highly motivated and parameter-efficient: Fisher Dynamic Expansion uses FIM-based metrics to avoid unnecessary adapter expansion, and Dictionary Replay replaces raw exemplars with a sparse embedding dictionary (solving privacy/memory issues), while the self-constructed benchmarks (capturing real-world semantic overlap) ensure rigorous evaluation.

**Weaknesses:**

1. It is recommended that the authors conduct a comparative analysis of the computational complexity and memory costs between the proposed ECA method and the compared baselines.

2. Only MoE-LoRA is included as a multi-modal IL baseline. Recent works in 2025 tailored for incremental vision-language task are omitted, making it difficult to fully assess ECA’s standing in the multi-modal IL landscape.

3. It is suggested that the authors discuss existing MLLM-based continual learning methods [1] [2] that have covered VQA and image captioning tasks, and explicitly elaborate on the significance of the proposed ECA method in comparison to these works within this paper.

[1] MCITlib: Multimodal Continual Instruction Tuning Library and Benchmark

[2] Continual LLaVA: Continual Instruction Tuning in Large Vision-Language Models

**Questions:**

See weakness

---

> ### Author Response · Authors · 2025-11-21
>
> Thank you for your constructive comments and suggestions! We have addressed all the comments and suggestions from the Reviewer and accordingly updated our manuscript highlighted in BLUE. We hope our responses below address your concerns. Please let us know if you have any additional concerns.
>
> Q1: Compare ECA’s computational complexity and memory usage with those of the baseline methods.
> -
> **A1:** Thank you for the suggestion. **In the revised version (highlighted in blue), we explicitly discuss efficiency in two places.**(i) In Sec. 5.1, we add a new Finding (4), showing that ECA uses almost the same number of trainable parameters as baselines with a single PA, yet consistently outperforms them and even methods with much larger parameter budgets. (ii) We also add a section about efficiency in Appendix H, where we compare trainable parameters, peak training/inference GPU memory, and inference throughput (tokens/s) on ToS-COCO Caption with the same dataset and GPU. The results are presented in the following **Table 1**.
>
> As shown in Table 1, ECA has essentially the same memory usage and throughput as the baselines with only a single PA, while Dual-Prompt, CODA-Prompt and MoE-LoRA are slower and require more memory. All methods freeze the BLIP-2 backbone and use the same training schedule, and ECA’s additional components (FeDEx and DR) only need to compute two extra gradient passes per task boundary plus light-weight dictionary updates. These results support our claim that **ECA improves continual learning performance without increasing trainable parameters, and hence maintains comparable computational complexity and memory cost.**.
>
> Table 1: Comparison of parameter, runtime, and memory efficiency on ToS-COCO Caption (BLIP-2 backbone: ViT-g/14 + OPT-2.7B, on NVIDIA A40).
> |Method|Trainable Params (M)|Peak GPU Mem (Train, GB)| GPU Mem (Infer, GB)| Inference Throughput (tokens/s) ↑|
> |-|-|-|-|-|
> |Vanilla (PA)|12.29|18.80|10.67|36.52|
> |Vanilla (Q-Former)|107.13|21.50|10.65|37.62
> |LwF|12.29|18.80|10.67|36.52|
> |EWC|12.29|18.80|10.67|36.52|
> |Dual-Prompt|14.30|19.47|10.70|32.82|
> |CODA-Prompt|15.41|19.55|10.70|34.77|
> |MoE-LoRA|98.84|21.37|11.02|36.37|
> |**ECA (ours)**|12.29|18.92|10.72|36.49|
>
> Q2,3: Discuss more recent multimodal continual-learning baselines, especially 2025 works, and clearly position ECA’s significance compared to them.
> -
> **A2:** In this paper, we focus on adapting to **ever-changing main topics in visual data**, while existing continual instruction tuning aims to understand different **instructions** without forgetting.
>
> Nevertheless, we discuss recent MLLM-based continual instruction-tuning methods, following the summary provided by the **very recent MCITlib library [1]**,  **in related work of the revision (highlighted in blue)**.
>
> More recently, several works on multimodal large language models introduced parameter-efficient fine-tuning (PEFT) approaches for **continual instruction tuning**. Specifically, Continual LLaVA[2] applied a Low-rank embedding pool, CoIN[3] and HiDe-LLaVA[4] leveraged MoE-LoRA in different ways, ModalPrompt[5] designed a dual-modality guided prompt tuning framework. However, these methods mainly aim at adapting models to evolving **textual instructions** rather than shifting **visual topics**.
>
> For this reason, we do not treat continual instruction-tuning methods as direct baselines in our setting.
>
> [1] MCITlib: Multimodal Continual Instruction Tuning Library and Benchmark. (Sep 2025, Arxiv)
>
> [2] Continual LLaVA: Continual Instruction Tuning in Large Vision-Language Models. (Nov 2024, Arxiv)
>
> [3] CoIN: A Benchmark of Continual Instruction Tuning for Multimodal Large Language Models. (Neurips 2024)
>
> [4] HiDe-LLaVA: Hierarchical Decoupling for Continual Instruction Tuning of Multimodal Large Language Model. (ACL 2025)
>
> [5] ModalPrompt: Dual-Modality Guided Prompt for Continual Learning of Large Multimodal Models. (EMNLP 2025)

---

> ### Author Response · Authors · 2025-11-26
> **Reminder to reviewer: Looking Forward to Your Further Comments and Feedback**
>
> Dear Reviewer sepo,
>
> We would like to thank you for taking the time to review our paper and for the insightful comments. We have addressed all the comments and suggestions you made. In particular, as suggested, we have **added an explicit efficiency comparison with baselines and expanded the related-work discussion on recent multimodal continual instruction tuning methods to clarify ECA’s position.** As we are approaching the midpoint of the discussion period, please kindly let us know if you have any additional concerns. We truly appreciate this opportunity to improve our work and shall be most grateful for any feedback you could give to us. If you do not have further questions, we are curious if you could consider raising our score.
>
> Thank you very much!

---

### Author Response · Authors · 2025-11-21

Dear Reviewers,

We sincerely thank you for your insightful comments and constructive suggestions. We have carefully revised the manuscript accordingly, **with all substantial changes highlighted in blue for ease of reference**, and provide detailed point-by-point responses below. We hope that our revisions and clarifications address your concerns, and we remain grateful for any further feedback.

---

### Meta-Review · Area_Chair_wnmH · 2026-01-06

**Summary:**

This work studies incremental learning when visual data shifts for open-ended image-to-text generation. The proposed Efficient Continual Alignment (ECA) only adapts the alignment module within pre-trained VLMs via three core components: a Mixture of Query (MoQ) module,  a Fisher Dynamic Expansion (FeDEx) and an embedding dictionary with Dictionary Replay (DR). In addition, four new benchmarks are developed to evaluate the performance under the target scenario.

**Reviewer Concerns:**

The initial concerns from reviewers include computational complexity and memory costs (Reviewer sepo, ZE8G, phKC) and the application to VLMs with different alignment modules (Reviewer DSk8, ZE8G, phKC). Besides, Reviewer sepo indicates that some related work is missing in the comparison/discussion. The rebuttal lists the comparison of trainable parameters, memory usage and inference throughput but training time is not reported. Moreover, the rebuttal conducts the experiments on LLaVA-v0 to show the applications of the proposed method on other VLM architectures. However, the LLaVA version is from 2023, which is not sufficient for demonstrating the SoTA performance. Therefore, the rebuttal may not address the major concerns thoroughly.

**Reviewer Scores:**

The initial scores for this work is mixed as 4,6,6,4. While the rebuttal did not fully address the concerns from reviewers with the score of 4, scores after rebuttal may not be increased significantly. The work can be further improved by incorporating the suggestions from reviewers with another round of revision.

---

### Decision · Program_Chairs · 2026-01-26

Reject